# ZERO-SHOT TEXT-BASED PERSONALIZED LOW-LIGHT IMAGE ENHANCEMENT WITH REFLECTANCE GUIDANCE

## ABSTRACT

Recent advances in zero-shot low-light image enhancement have largely benefited from the deep image priors encoded in network architectures. However, these models require optimization from scratch for each image and cannot provide personalized results based on user preferences. In this paper, we propose a training-free zero-shot personalized low-light image enhancement model that integrates Retinex domain knowledge into a pre-trained diffusion model, enabling style personalization based on user preferences specified through text instructions. Our contributions are as follows: First, we incorporate the total variation optimization into a single Gaussian convolutional layer, enabling zero-shot Retinex decomposition. Second, we introduce the Contrastive Language-Image Pretraining (CLIP) model into the reflectance-conditioned sampling process of Denoising Diffusion Implicit Models (DDIM), guiding the enhancement according to user-provided text instructions. Third, to ensure consistency in content and structure, we employ patch-wise DDIM inversion to find the initial noise vector and use the reflectance as a condition during the reverse sampling process. Our proposed model, RetinexGDP, supports any image size and produces noise-suppressed results without imposing extra noise constraints. Extensive experiments across nine low-light image datasets show that RetinexGDP achieves performance comparable to state-of-the-art models.

## 1 INTRODUCTION

Low-light image enhancement (LLIE) algorithms aim to restore details hidden in dark areas while preserving color accuracy and naturalness. These algorithms not only enhance the human visual experience but also improve the performance of high-level computer vision tasks Liu et al. (2021a). Retinex-based LLIE models Ren et al. (2018); Xu et al. (2020); Yang et al. (2021); Zhang et al. (2019) have gained attention by incorporating Retinex theory into deep neural networks. According to Retinex theory, an image can be modeled as the product of reflectance and illumination: $S = R \odot I$, where $\odot$ denotes the Hadamard product. Thus, Retinex decomposition, which involves estimating these components, is a central problem for these methods. With the derived reflectance and illumination, models such as those proposed by Hao et al. (2020); Liang et al. (2022), enhance images by adjusting the illumination and denoising the reflectance before recombining the two components. Alternatively, some methods separate illumination from reflectance to achieve enhancement Guo et al. (2017); Zhao et al. (2021). These methods focus on adjusting illumination but pay less attention to the style of the reflectance, which limits their ability to offer personalized image enhancement based on user preferences.

Personalized low-light enhancement (PLIE) is able to enhance low-light image according to diverse user preference, however, it is less explored compared to general LLIE. Existing PLIE models either require users to select preferred images to represent desired styles Kim et al. (2020) or inject user profiles into the network for personalization Bianco et al. (2020). Additionally, these models apply a single style preference vector to all images, limiting the diversity of styles. Recently, masked style modeling is applied to image enhancement, achieving content-aware personalization Kosugi & Yamasaki (2024). Despite the success of these models, a common problem is the inability to enhance unseen images in a preferred style that does not exist in the database. Although researchers

can enrich the database by collecting more diverse preference images, this would require retraining or fine-tuning the network.

In this work, we propose a zero-shot text-based personalized low-light image enhancement model, RetinexGDP, that requires no training or parameter fine-tuning. This model allows users to specify enhancement style preferences via text instructions. The core idea behind RetinexGDP is to integrate Retinex domain knowledge into a pre-trained diffusion model while leveraging Contrastive Language-Image Pretraining (CLIP) Radford et al. (2021) to guide the Generative Diffusion Prior (GDP) Fei et al. (2023). To achieve zero-shot Retinex decomposition, we take the total variation optimization as a layer and incorporate it into a single Gaussian convolutional layer to estimate the illumination map. Instead of designing deep network, our RetinexGDP requires only a single convolutional layer for illumination map estimation. We then compute the corrected reflectance in spatial domain and consider it as the initial enhanced result. At the stage of personalized enhancement, we first find the initial noise vector of the corrected reflectance map by patch-wise DDIM inversion. To maintain the content and structure consistency of diffusion model, the corrected reflectance is used as a conditional input during the reverse sampling process of the diffusion model, where the sampling takes the initial noise vector as starting point and is guided by directional CLIP loss, steering the enhancement toward the style specified by the user through text. Guided by content and style loss, RetinexGDP enhances the low-light image according to user-specified preferences without the need for retraining or fine-tuning from scratch. RetinexGDP is adaptable to images of any input size. Experiments have shown that, despite the absence of additional noise constraints, the enhancement results generated by our model exhibit good performance in noise suppression.

In summary, our main **contributions** are summarized as follows. (I) We propose a zero-shot text-based PLIE model named RetinexGDP, enabling flexible enhancement guided by user preferences specified via text instructions, without the need for additional training or external images. (II) We incorporate the edge-aware property of total variation optimization into a single Gaussian convolutional layer, aiming to perform zero-shot Retinex decomposition. (III) We employ patch-wise DDIM inversion to generate the initial noise vector of corrected refletance and take the corrected refletance as condition of DDIM reverse process, maintaining the image content and structures.

## 2 RELATED WORK

**Zero-shot Low-light Image Enhancement.** While deep learning-based methods Zhang et al. (2022); Huang et al. (2023); Zhang et al. (2023); Xu et al. (2023), particularly those combined with the Retinex model Yang et al. (2021); Wu et al. (2022); Xu et al. (2022); Fu et al. (2023), have demonstrated superior performance compared to traditional optimization-based techniques Fu et al. (2016); Guo et al. (2017); Li et al. (2018); Xu et al. (2020), zero-shot low-light image enhancement (LLIE) remains relatively underexplored. Zero_DCE Guo et al. (2020) and its extension Zero_DCE++ Li et al. (2021) propose predicting higher-order curves through iterative self-application, independent of paired and unpaired external data. The Retinex decomposition is transformed into a generative problem in Zhao et al. (2021); Liang et al. (2022), where combined deep image priors (DIP) Ulyanov et al. (2020); Gandelsman et al. (2019) are applied to generate latent Retinex components without the need for any external training dataset. Those methods perform zero-shot Retinex decomposition. However, they rely on deep priors encoded in the network structure, with network parameters randomly initialized and no data priors utilized. Consequently, their networks require optimization from scratch for each entry. In this work, we incorporate the edge-preserving property of total variation optimization into a single Gaussian convolutional layer, without requirement of very deep network.

**Personalized Low-light Image Enhancement.** PLIE aims to enhance low-light images based on user preference. Traditional PLIE methods enhance image based on the user's preference through simple gamma correction or S-curve Kapoor et al. (2014). With the advent of deep learning, convolutional neural networks (CNNs) have been employed to extract preference vectors, which are then used for personalized enhancement Kim et al. (2020). Later, the user profile with feature vectors are integrated to enhance images Bianco et al. (2020). Recently, a style-aware model using a style encoder that learns image embeddings is proposed to map preferred styles to latent codes Song et al. (2021). However, these approaches apply the same preference vector to all images, without considering the content of the preferred images. More recently, masked style modeling is adopted for

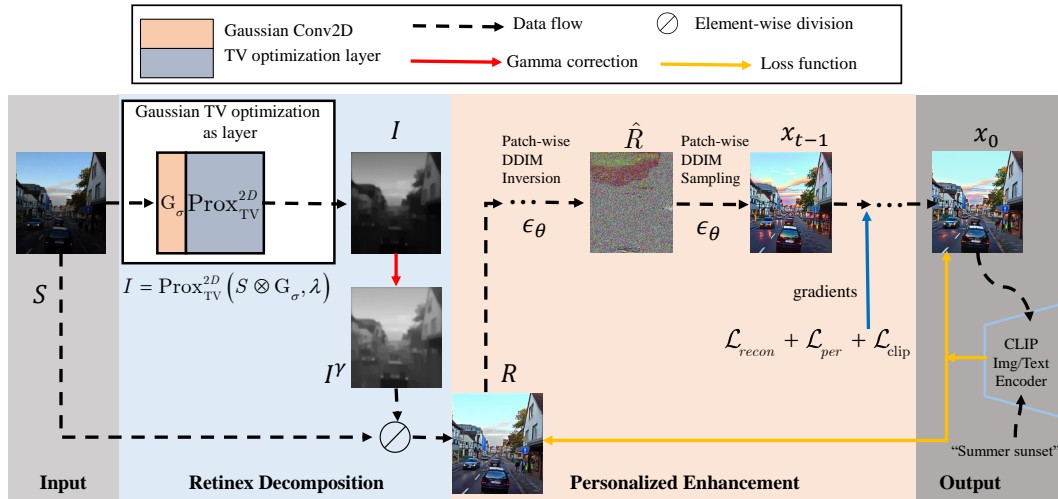

Figure 1: Overview of our RetinexGDP. $S$ and $R$ is the input low-light image and corrected reflectance, respectively. $\hat{R}$ denotes the initial noise vector. $\gamma$ is the gamma factor. $x_0$ is the enhanced image. $\mathcal{L}_{recon}$, $\mathcal{L}_{per}$, and $\mathcal{L}_{clip}$ denote the reconstruction loss, perceptual loss and style loss, respectively.

content-aware personalized image enhancement Kosugi & Yamasaki (2024). This method employs a transformer encoder to predict style embeddings for unseen images by considering similar content in the preferred image dataset. Nevertheless, the aforementioned methods are either restricted to the style of the collected preferred images or require retraining on new dataset. In our work, the preferred style of enhancement is specified by user-provided text and our personlization requires no training.

**Diffusion Model Based Low-light Image Enhancement.** In recent years, diffusion probabilistic models have achieved significant success in image generation and manipulation. However, challenges remain in effectively integrating low-light images as conditions and designing appropriate diffusion models. To address this, CLE-Diffusion Yin et al. (2023) concatenates the color map with the low-light input to preserve color information. Diff-Retinex Yi et al. (2023) introduces a three-stage framework where two diffusion models adjust reflectance and illumination components. ReCo-Diff Wu et al. (2023) combines the low-light input with the generated image and performs Retinex decomposition at each time step. Other approaches modify the diffusion process itself for low-light enhancement Wang et al. (2023); Jiang et al. (2023). A recent work Reti-diff He et al. (2023) introduces the Retinex priors to latent diffusion, however, it requires additional networks or retraining on external datasets. In contrast, recent works Fei et al. (2023); Jiang et al. (2024); Lv et al. (2024) leverage a pretrained Denoising Diffusion Probabilistic Model (DDPM) as a generative prior to optimize the reverse sampling process, enabling training-free image enhancement. Building on these advancements, we incorporate the Retinex domain knowledge into generative diffusion prior and propose a zero-shot text-based personalized low-light enhancement model.

## 3 METHOD

The goal of this work is to develop a fully text-driven, training-free PLIE model. Our RetinexGDP operates in two stages: zero-shot Retinex decomposition and text-guided personalized enhancement, both of which are training-free. The overview of our RetinexGDP is shown in Fig.1.

### 3.1 ZERO-SHOT RETINEX DECOMPOSITION

Previous Retinex-based LLIE works Zhao et al. (2021); Liang et al. (2022) have leveraged DIP to perform zero-shot Retinex decomposition, exploiting DIP's inductive bias for Retinex component

generation. In practice, these models require additional hand-crafted priors to ensure piecewise smoothness, which is important to illumination estimation. To eliminate the need for such priors, the edge-aware smoothness properties in a bilateral grid and an encoder-like DIP model is combined to estimate the illumination Zhao et al. (2024). Despite its success in illumination estimation, this approach still necessitates a DIP network comprising many convolutional blocks.

**Is it possible that illumination estimation be performed with a single convolutional layer?** The main challenge is how to embed inductive bias into a convolutional layer to guide optimization towards a piecewise smooth solution. While convolutional blocks possess inductive biases such as locality and translation equivariance and can be used to smooth the details in an image, they are insufficient for decomposing an image into piecewise smooth illumination. In contrast, the TV optimization can be applied to preserve the image edges, as illustrated in Fig.11 in Appendix A.2. Hence, if the edge-preserving smoothness properties of TV optimization can be incorporated into a single convolutional layer, we do not need a deep network for illumination estimation.

**How to incorporate the edge-preserving properties into a single convolutional layer?** Total variation (TV) is an effective regularizer that has been widely used as a smoothness regularization term in denoising. Considering an input image $\boldsymbol{X}$ and output $\boldsymbol{Y} \in \mathbb{R}^{M \times N}$, the TV optimization can be defined as:

$$\arg \min_{\boldsymbol{Y}} \frac{1}{2} \|\boldsymbol{Y} - \boldsymbol{X}\|_F^2 + \lambda \|D\boldsymbol{Y}\|_1, \tag{1}$$

where $D = [D_x^T, D_y^T]^T$ denote the first order forward finite-difference matrix along the row and column directions respectively, and $\lambda$ indicates the balance parameter for controlling the strength of regularization. $\| \cdot \|_F$ and $\| \cdot \|_1$ denote Frobenius norm and $L_1$ norm, respectively. Inspired by Yeh et al. (2022), which reports that total variation optimization as a layer (TV layer) provides effective piece-wise properties and can be used to inject specific inductive bias to the deep network during both training and inference. Given an input feature map $\boldsymbol{X} \in \mathbb{R}^{C \times H \times W}$ the TV layer outputs a tensor $\boldsymbol{Y}$ of the same size. This layer computes the TV proximity operator independently for each channel. The forward operation can be summarized as:

$$\boldsymbol{Y}_c = \mathrm{Prox}_{\mathrm{TV}}^{2D} \left( \boldsymbol{X}_c, \lambda \right), \tag{2}$$

where $\mathrm{Prox}_{\mathrm{TV}}^{2D}(\cdot)$ denotes the TV proximity operator, and $c$ represents $c$-th channel of a tensor.

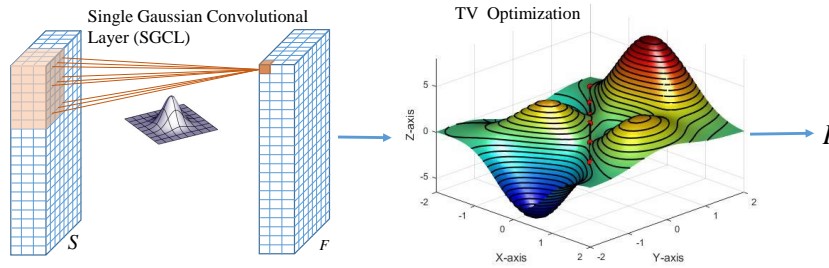

Figure 2: Illustration of TV optimization as a layer for zero-shot illumination estimation.

We therefore incorporate this TV proximity operator as a layer into a single convolutional layer, as shown in Fig.2. Previous study seamlessly integrated the differentiable TV layer into a deep neural network for training purposes Yeh et al. (2022), in which both the convolutional kernel and the balance parameters are trainable within this setup. However, our objective is to develop a training-free model, which means our illumination estimation model involves only the forward process without a backward process for parameter updates. This presents a problem as the parameters of the convolutional layer are random values, leading to wrong illumination estimation, as shown in Fig. 3(a). We observe distinct differences in the illumination maps produced by the vanilla convolutional TV layer across the three experiments. Inaccurate or inconsistent illumination estimation can result in incorrect image enhancement outcomes.

To mitigate this issue, we adopt a strategy wherein we replace the vanilla convolutional kernel with a *Gaussian* kernel. The coefficients of this Gaussian filter are sampled from a normal distribution

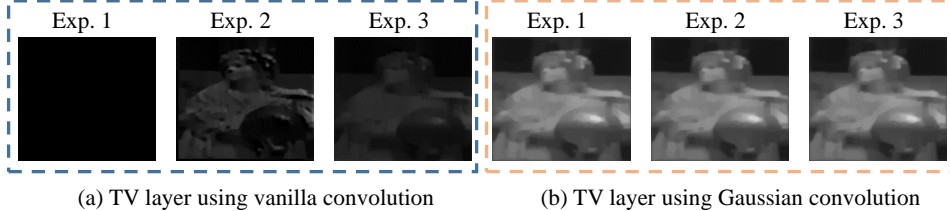

(a) TV layer using vanilla convolution      (b) TV layer using Gaussian convolution

Figure 3: Comparison of illumination estimation when TV layer uses vanilla convolution kernel and Gaussian kernel.

with mean 0 and variance $\sigma^2$. Hence, we define our Gaussian TV layer for illumination estimation by:

$$\boldsymbol{I}_c = \mathrm{Prox}_{\mathrm{TV}}^{2D}\left(\boldsymbol{S}_c \otimes \mathrm{G}_\sigma, \lambda\right), \tag{3}$$

where $\otimes$ denotes the convolution operation, and $\mathrm{G}_\sigma$ denotes the Gaussian filter. $\boldsymbol{S}$ is the input image. $\lambda$ is the balance parameter in Eq.1. Consequently, the Fourier spectrum of our Gaussian TV layer can be manipulated by adjusting the value of $\sigma$. With a predefined $\sigma$ and fixed kernel size, the parameters of Gaussian TV layer are deterministic, leading to consistent illumination estimation. Despite the absence of a training process, the Gaussian TV layer is capable of generating a piece-wise smooth illumination, as demonstrated in Fig. 3(b).

This illumination estimation method offers several advantages. First, it requires no external images for training and relies solely on the input single image itself, thus overcoming challenges associated with dataset collection. Second, it necessitates no additional hand-crafted priors or loss functions to drive optimization, thereby streamlining our illumination estimation procedure. Third, our Gaussian TV layer requires only one convolutional layer.

Once the illumination is obtained, we directly compute the reflectance in spatial domain. Since a corrected reflectance can be regarded as initial enhanced image Guo et al. (2017); Zhao et al. (2024), we consider the corrected reflectance as initial enhanced image. However, the reflectance obtained by uncorrected illumination cannot be considered as a initial enhanced image. We therefore use Gamma correction to obtain the corrected illumination $\boldsymbol{I}^\gamma$, where $\gamma$ is the Gamma correction factor. Then we compute the corrected reflectance according to Retinex model:

$$\boldsymbol{R} = \boldsymbol{S}_c \oslash \boldsymbol{I}^\gamma, c \in \{R, G, B\}, \tag{4}$$

where $\oslash$ is spatial division operation and $c$ indicates the $c$-th channel in RGB color space.

A sample of Retinex decomposition in our RetinexGDP is shown in Figure 4. Notably, the illumination map exhibits piece-wise smooth characteristic, ensuring that rich details and color information are entirely preserved in the reflectance. The corrected reflectance, as the initial enhanced image, is utilized in the subsequent stage of personalized enhancement.

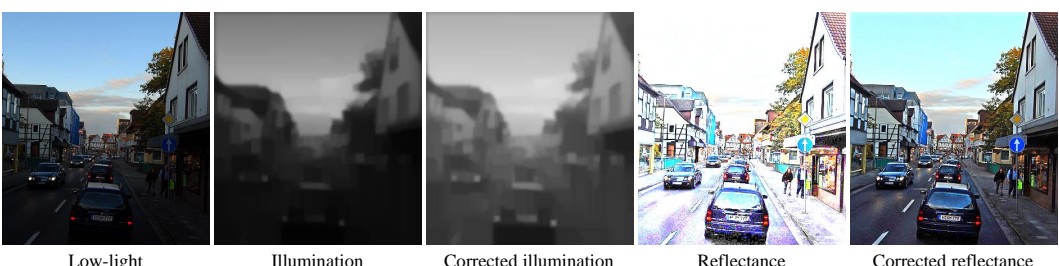

Low-light      Illumination      Corrected illumination      Reflectance      Corrected reflectance

Figure 4: The results of Retinex decomposition.

## 3.2 TEXT-BASED PERSONALIZED LOW-LIGHT ENHANCEMENT

**Finding initial noise vector of corrected reflectance using patch-wise DDIM inversion.** The initial noise vector plays a crucial role in maintaining the fidelity of the generated original image.

The Illustration of finding the initial noise vector of corrected reflectance is shown in Fig.5. DDIM inversion aims to deterministically noising the corrected reflectance $\boldsymbol{R}$ to obtain a noise vector $\hat{\boldsymbol{R}}$, which differs from the pure noise strategy of GDP Fei et al. (2023). In the inversion process, the corrected reflectance $\boldsymbol{R}$ is first divided into $M$ overlapping patches, which are cropped with a stride of $p$. For each patch, we obtain the noised intermediate result. At each time step $t$, the diffusion model computes the mean $\mu^m$ and variance $\Sigma^m$ of Gaussian noise for each patch. During the diffusion process, these values are iteratively shifted to reflect the overall mean and variance for the entire image.

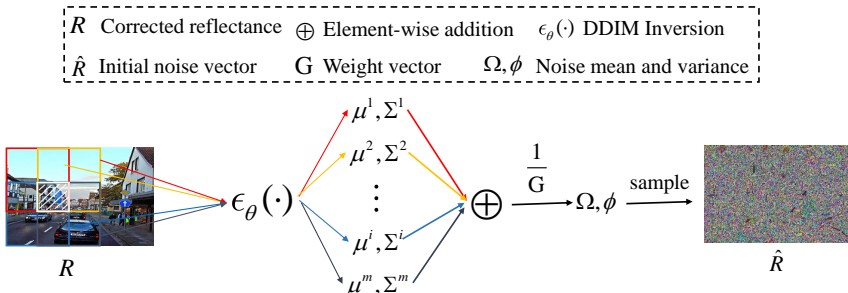

Figure 5: Illustration of finding the initial noise vector of corrected reflectance using patch-wise DDIM inversion.

Since the patches overlap, the overlapping areas are computed multiple times. Therefore, it is crucial to record the number of times these overlapping areas are noised to ensure accurate calculations. A binary patch mask $\mathbf{P}^m$ is used to locate the overlapping areas, and weight vector $\mathbf{G}$ is used to maintain a count of how often each pixel in the overlapped areas is included in a patch: $\mathbf{G} = \mathbf{G} + \mathbf{P}^m$, where $m$ indicate $m$-th patch. The final mean and variance for the whole image at each time step are then obtained by computing the weighted average of $\Omega_t$ and the variance vector $\phi_t$: $\Omega_t = \Omega_t \oslash \mathbf{G}$, $\phi_t = \phi_t \oslash \mathbf{G}$, where $\oslash$ represents the element-wise division. This approach ensures that the contributions from overlapping patches are aggregated, preserving the consistency of the image's structure and content during the subsequent sampling process.

**Text-based sampling conditioned on reflectance.** The initial noise vector $\hat{\boldsymbol{R}}$ produced through DDIM inversion, rather than a pure Gaussian noise, is taken as the starting point in the reverse denoising step, to maintain the data consistency. To provide a high-quality and more reliable condition for the guided diffusion model Dhariwal & Nichol (2021), we additionally use the corrected reflectance $\boldsymbol{R}$ as the condition in the reverse denoising process. By conditioning on the corrected reflectance, our method ensures that the generated outputs maintain the original image's structure and texture while enhancing low-light image.

We now describe how to perform reverse sampling from $\hat{\boldsymbol{R}}$. If the reverse denoising distribution $P(x_{t-1} \mid x_t)$ is adopted to a conditional distribution $P(x_{t-1} \mid x_t, \boldsymbol{R})$, $P(\boldsymbol{R} \mid x_t)$ can be a probability that $x_t$ will be denoised to a high-quality image consistent to $\boldsymbol{R}$, and according to Dhariwal & Nichol (2021), its heuristic approximation is formulated as

$$P(\boldsymbol{R} \mid \boldsymbol{x}_t) = \frac{1}{Z} \exp\left(-\left[\lambda_1 \mathcal{L}_c(\boldsymbol{x}_t, \boldsymbol{R}) + \lambda_2 \mathcal{L}_s(\boldsymbol{x}_t, \boldsymbol{R}, \mathrm{d})\right]\right). \quad (5)$$

where $Z$ is a normalizing factor, $\mathcal{L}_c$ and $\mathcal{L}_s$ indicate the image content and style distance metrics, respectively. $\mathcal{L}_c$ is consist of reconstruction loss $\mathcal{L}_{recon}$ and perceptual loss $\mathcal{L}_{per}$. It ensures that the fine details and structures in the reflectance map are preserved. We use pretrained CLIP model to style distance $\mathcal{L}_s$. $\mathcal{L}_s$ ensures the generated image matches the desired style, as guided by the user's text prompt. d is the text instruction of enhancement style, and $\lambda_1$ and $\lambda_2$ are scaling factors for controlling the magnitude of guidance. In this way, the conditional transition $P(\boldsymbol{x}_{t-1} \mid \boldsymbol{x}_t, \boldsymbol{R})$ an be approximately obtained through the unconditional transition $P(\boldsymbol{x}_{t-1} \mid \boldsymbol{x}_t)$ by shifting the mean of the unconditional distribution: $\mu = \mu + \Sigma \nabla_{\boldsymbol{x}_t} \log P(\boldsymbol{R} \mid \boldsymbol{x}_t)$, where

$$\nabla_{\boldsymbol{x}_t} \log P(\boldsymbol{R} \mid \boldsymbol{x}_t) = -\lambda_1 \nabla_{\boldsymbol{x}_t} \mathcal{L}_c(\boldsymbol{x}_t, \boldsymbol{R}) - \lambda_2 \nabla_{\boldsymbol{x}_t} \mathcal{L}_s(\boldsymbol{x}_t, \boldsymbol{R}, \mathrm{d}) \quad (6)$$

Therefore, we can add guidance on the generation process by direct adding the mean shift (the gradients of loss function for content and style) to the intermediate denoised image. the gradients are

added to the denoised image at each time step, which is actualy shifting the mean of the unconditional distrubution. Shifting mean enables the generated images during DDIM sampling to be closer to the distribution of personalized augmented images while maintaining consistent content.

Instead of directly compute the gradients between the intermediate denoised image and the initial reflectance, to avoid the regression-to-the-mean effects, we compute the gradients of the linear combination of them: $\tilde{x}_{0,t-1} = \eta x_{0,t-1} + (1 - \eta)x_{0,t}$, where $\eta = \sqrt{1 - \bar{\alpha}_t}$. In practice, the condition is a linear combination of $\boldsymbol{R}$ and $\tilde{\boldsymbol{R}}$. $\tilde{\boldsymbol{R}}$ is sampled by $\tilde{\boldsymbol{R}}_t = \sqrt{\bar{\alpha}_t}\boldsymbol{R}_0 + \sqrt{1 - \bar{\alpha}_t}\epsilon$, and $\boldsymbol{R}_t = \eta\boldsymbol{R}_0 + (1 - \eta)\tilde{\boldsymbol{R}}_t$. We investigate the impact of $\eta$ on enhancement performance in Appendix.A.5.1. By finetuning the scale factors of loss functions while specifying the preference with text prompt, we can control the guidance.

### 3.2.1 LOSS FUNCTIONS

The loss functions used in the framework contain two parts: content guidance $\mathcal{L}_c$ and style guidance $\mathcal{L}_s$. To preserve structure and texture consistency between the reconstructed image and the initial enhanced image, $\mathcal{L}_c$ is consist of reconstruction loss $\mathcal{L}_{recon}$ and perceptual loss $\mathcal{L}_{per}$.

**Reconstruction loss.** We aim to maintain the structure and texture consistency between the enhanced image and the input image, except for noise and illumination. To ensure this similarity, we minimize the Mean Squared Error (MSE) between the corrected reflectance, i.e., the initial enhanced image, and the output.

$$\mathcal{L}_{recon} = \|\tilde{x}_{0,t} - \boldsymbol{R}_t\|_2^2, \tag{7}$$

where $\boldsymbol{R}_t$ and $\tilde{x}_{0,t}$ denote $t$-th sampled corrected reflectance and the linear combination of intermediate output, respectively.

**Perceptual loss.** Additionally, to improve visual sharpness of the enhanced image, we adopt the perceptual loss defined by the similarity on the extracted feature maps from 2 layers of the pretrained VGG19 network.

$$\mathcal{L}_{per} = \|\phi_k(\tilde{x}_{0,t}) - \phi_k(\boldsymbol{R}_t)\|_2^2, \tag{8}$$

where $\phi_k(\cdot)$ denotes the extracted feature using $k$-th layer of the pre-trained VGG19 network.

**Style loss.** We use directional CLIP loss Gal et al. (2022) to align the direction between text instructions and image pairs in the CLIP embedding space. However, we find that the source text prompt does not benefit content consistency. In our experiment, the source text prompt (if used) describes the style of the corrected reflectance, while the target text prompt describes the desired style. However, there appears to be a misalignment between natural language descriptions and the reflectance component. Therefore, we modify the directional CLIP loss by removing the source prompt:

$$\mathcal{L}_{\text{clip}}\left(\tilde{x}_{0,t}, \boldsymbol{R}_t, p_{\text{target}}\right) = 1 - \frac{\Delta I \cdot E_{txt}\left(p_{\text{target}}\right)}{\|\Delta I\|\|E_{txt}\left(p_{\text{target}}\right)\|}, \tag{9}$$

where $\Delta I = E_{img}\left(\tilde{x}_{0,t}\right) - E_{img}\left(R_t\right)$. $E_{img}$ and $E_{txt}$ are CLIP's image embedding and text embedding obtained by image encoder $E_{img}$ and text encoder $E_{txt}$, respectively. Users can specify their preferred style using text prompts, denoted as $p_{\text{target}}$, such as "summer sunset" or "bright daylight," for personalized enhancement.

## 4 EXPERIMENTS

**Experimental setups.** The kernel size of the Gaussian TV layer is 7, with stride 1, and the value of $\sigma$ is set to 0.5. We adjust the balance parameter $\lambda$ in the Gaussian TV layer to 30, and perform a single iteration. The Gamma factor $\gamma$ is set to 0.5. We use the pretrained unconditional guided diffusion model as our backbone Dhariwal & Nichol (2021). For the pretrained CLIP model, we adopt "ViT-B-32". The scaling factor for $\mathcal{L}_{recon}$, $\mathcal{L}_{per}$ and $\mathcal{L}_{\text{clip}}$ are set to 5000, 100 and 7000, respectively. The total number of time steps, $T$, is 50, and we space the step size from $T$ to $T'$, where $T'$ is set to 25 in our experiment. Additionally, we employ patch strategy in both the inversion and reverse processes, with the patch size set to 256. Our experiments are conducted on a single TITAN X GPU.

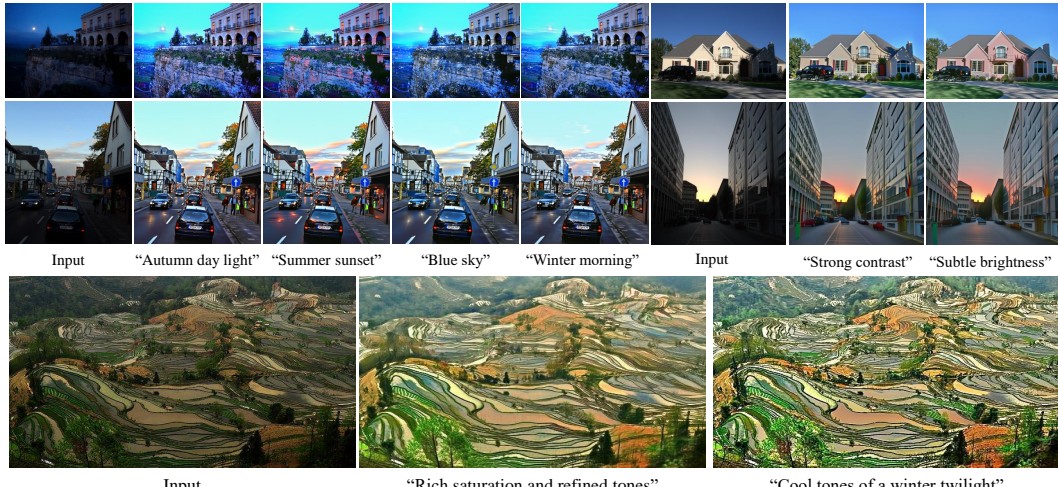

Figure 6: The enhanced results guided with different text instructions. Zoom in for better viewing.

**Baseline Implementations.** The proposed model is compared against with 10 state-of-the-art LLIE models: Training-based models: Zero_DCE Guo et al. (2020), SNRXu et al. (2022), DCCNet Zhang et al. (2022), UHDFour Li et al. (2023), URetinexNetWu et al. (2022), DiffLLJiang et al. (2023), CLIPLITLiang et al. (2023). Training-free models: RetinexDIP Zhao et al. (2021), DRP Liang et al. (2022), NeuralBRZhao et al. (2024).

**Metrics.** Three no-reference image assessment metrics (NIQE, CPCQIGu et al. (2017) and NIQMC Gu et al. (2018)) are utilized in the paper. Since LOL and VELOL datasets have paired images, we therefore use PSNR and SSIM metrics to evaluate our model.

**Datasets.** We evaluate our model on 9 public datasets including 224 real images (DICMLee et al. (2012), ExDarkLoh & Chan (2019), FusionMa et al. (2015), LIMEGuo et al. (2017), NASA, NPEWang et al. (2013), VV, LOL dataset Wei et al. (2018) and VELOL datasetLiu et al. (2021b)).

## 4.1 PERSONALIZED LOW-LIGHT IMAGE ENHANCEMENT.

We enhance low-light image under different text guidance that describes various styles. The enhanced results guided with different text instructions are given in Fig.6. It can be observed that, with the input nighttime or insufficient light image, our RetinexGDP successfully enhances low-light image according to user preferred style specified by text, and both the structures and textures are preserved well. For example, when the preference specified with "summer sunset", in the enhanced image, the leaves of the trees turn green, the evening glow on the horizon displays an orange hue. When the preference specified with "Blue light", giving the entire image a cold atmosphere. In addition, without denoising post-processing, our RetinexGDP does not amplify the noises, leading to the enhanced results more pleasing.

## 4.2 GENERAL LOW-LIGHT IMAGE ENHANCEMENT

**Qualitative assessments.** We present enhanced results obtained from real low-light datasets, as shown in Fig. 7 and 8. From Figure 7, our RetinexGDP not only restores details hidden in the dark, but also effectively removes noise. In contrast to other supervised models, such as CLIP-LIT and DiffLL, our enhanced results have higher image contrast and vivid color, without over-enhancing relatively bright areas, as shown in Fig.8.

**Quantitative assessments.** We quantitatively assess the performance of our RetinexGDP and competitive methods across 9 datasets, as shown in Table 1. While our model does not achieve the top performance across all datasets, it consistently exhibits competitive results on several datasets, showcasing its versatility and robustness in various real-world scenarios. For instance, considering

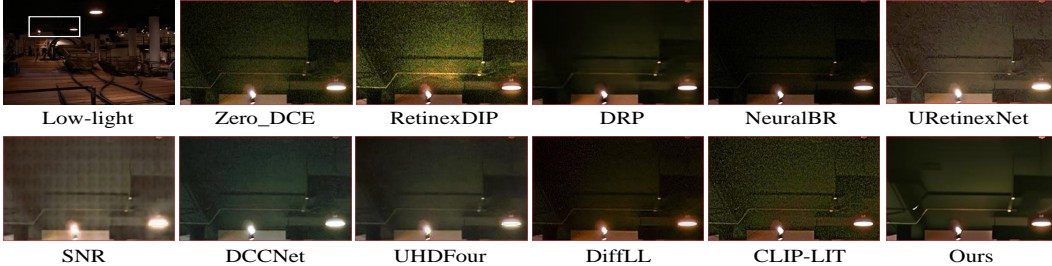

Figure 7: Denoising performance comparison with the state-of-the-art LLIE methods on real dataset.

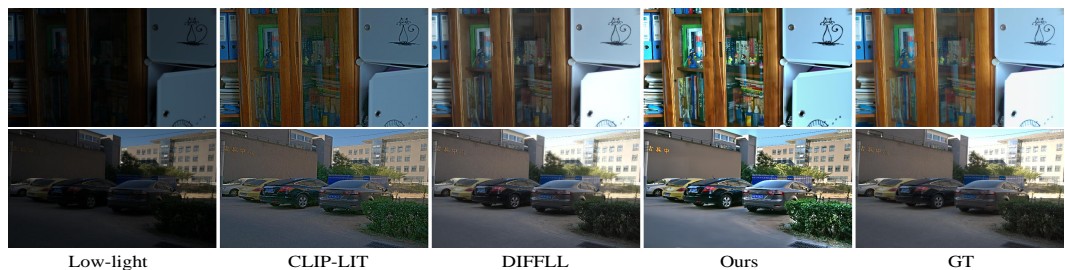

Figure 8: Image contrast visual comparison with the state-of-the-art LLIE methods on LOL and VELOL dataset.

Table 1: Average NIQE ↓, NIQMC ↑, and CPCQI ↑ scores across seven datasets. Training-free models are highlighted in gray. The best image quality is marked in red (1st), green (2nd), and blue (3rd). A '-' indicates results are unavailable due to memory issues.

| Method | DICM | | | ExDark | | | Fusion | | | LIME | | | Nasa | | | NPEA | | | VV | | |
|---|---|---|---|---|---|---|---|---|---|---|---|---|---|---|---|---|---|---|---|---|---|
| | N↓ | M↑ | C↑ | N↓ | M↑ | C↑ | N↓ | M↑ | C↑ | N↓ | M↑ | C↑ | N↓ | M↑ | C↑ | N↓ | M↑ | C↑ | N↓ | M↑ | C↑ |
| URetinexNet | 3.50 | 5.17 | 0.72 | 3.74 | 5.18 | 0.87 | 3.90 | 5.07 | 0.85 | 4.33 | 5.02 | 0.90 | 3.27 | 5.14 | 0.84 | 4.08 | 5.00 | 0.85 | 3.03 | 5.14 | 0.79 |
| SNR | 4.16 | 5.17 | 0.63 | 4.29 | 5.29 | 0.68 | 4.93 | 5.00 | 0.64 | 5.69 | 5.23 | 0.72 | 5.23 | 5.28 | 0.57 | 4.15 | 4.98 | 0.67 | 8.77 | 4.98 | 0.45 |
| DCCNet | 3.29 | 5.02 | 0.75 | 3.75 | 5.07 | 0.74 | 4.29 | 4.80 | 0.77 | 4.26 | 5.04 | 0.88 | 3.18 | 5.16 | 0.87 | 3.50 | 4.74 | 0.75 | 3.56 | 5.11 | 0.76 |
| UHDFour | 3.46 | 5.09 | 0.75 | 3.90 | 5.00 | 0.71 | 4.38 | 4.85 | 0.72 | 4.55 | 4.97 | 0.85 | 3.33 | 5.39 | 0.85 | 3.62 | 4.94 | 0.74 | - | - | - |
| DiffusionLL | 2.93 | 5.22 | 0.77 | 3.27 | 5.04 | 0.79 | 3.30 | 5.19 | 0.80 | 3.58 | 4.92 | 0.95 | 2.81 | 5.33 | 0.82 | 3.24 | 5.00 | 0.81 | 2.92 | 5.36 | 0.88 |
| CLIPLIT | 3.01 | 5.05 | 0.84 | 3.63 | 4.84 | 1.04 | 3.74 | 5.09 | 1.00 | 3.99 | 5.09 | 1.00 | 3.16 | 5.19 | 1.04 | 3.71 | 4.97 | 0.98 | 3.02 | 5.20 | 1.00 |
| Zero_DCE | 2.83 | 5.12 | 0.82 | 3.54 | 4.96 | 0.97 | 3.58 | 5.21 | 0.91 | 3.76 | 4.84 | 1.06 | 3.57 | 5.09 | 0.87 | 2.97 | 4.89 | 0.92 | 3.21 | 5.40 | 0.89 |
| LightenDiffusion | 3.39 | 5.23 | 0.90 | 3.34 | 5.14 | 0.80 | 3.43 | 5.21 | 0.78 | 4.04 | 5.10 | 0.94 | 3.08 | 5.48 | 0.89 | 3.03 | 5.14 | 0.79 | 3.58 | 5.38 | 0.77 |
| FourierDiff | 3.97 | 4.94 | 0.86 | 3.80 | 5.17 | 0.82 | 4.33 | 4.66 | 0.83 | 4.21 | 5.16 | 0.97 | 3.37 | 4.75 | 0.85 | 3.72 | 5.06 | 0.84 | 3.36 | 4.75 | 0.85 |
| RetinexDIP | 3.37 | 5.13 | 0.86 | 3.74 | 4.86 | 1.13 | 3.40 | 5.33 | 1.05 | 3.82 | 4.88 | 1.16 | 3.58 | 5.41 | 1.02 | 3.01 | 5.15 | 1.04 | 2.48 | 5.45 | 1.06 |
| DRP | 4.68 | 5.24 | - | 4.79 | 5.17 | - | 5.71 | 5.28 | - | 5.99 | 5.21 | - | 4.30 | 5.62 | - | 5.29 | 5.37 | - | 8.80 | 5.45 | - |
| NeuralBR | 3.39 | 5.29 | 0.88 | 3.79 | 4.72 | 1.08 | 3.41 | 5.23 | 1.05 | 3.74 | 5.03 | 1.14 | 2.97 | 5.19 | 1.04 | 3.72 | 5.15 | 1.05 | 3.21 | 5.40 | 0.89 |
| RetinexGDP | 4.02 | 5.12 | 0.84 | 4.80 | 4.97 | 0.81 | 5.22 | 5.27 | 0.86 | 5.54 | 5.06 | 0.94 | 4.11 | 5.48 | 0.91 | 4.21 | 5.38 | 0.75 | 4.10 | 5.26 | 0.74 |

our NIQMC score on the NPEA dataset, as detailed in Table 1, our RetinexGDP model achieves the highest score.

Furthermore, we specifically compare our RetinexGDP with models trained on the LOL dataset, as shown in Table 2. Compared to training-based models, such as CLIP-LIT, our model achieves significantly higher PSNR scores: 26.39% on LOL and 8.7% on VELOL. Similarly, against training-free models such as RetinexDIP, RetinexGDP delivers substantial improvements, with 82.3% higher PSNR on LOL and 48.9% on VELOL. In contrast, without being specifically trained on the LOL or VELOL datasets, RetinexGDP achieves superior performance.

## 4.3 ABLATION STUDY

**Loss function and text prompts**. We conduct an ablation study on the loss function and text specified preference, as shown in Table 3. We observe that using only content guidance ($\mathcal{L}_{recon}$ and $\mathcal{L}_{per}$) yields the enhanced image with better image quality, while the addition of text instruction may result in a slight drop in performance. More visual results are given in Appendix A.5.2.

Table 2: Average PSNR↑/SSIM↑/NIQE↓/NIQMC↑/CPCQI↑ on LOL and VELOL.Training-free models are highlighted in gray.

| Method | LOL | | | | | VELOL | | | | |
|---|---|---|---|---|---|---|---|---|---|---|
| | P↑ | S↑ | N↓ | M↑ | C↑ | P↑ | S↑ | N↓ | M↑ | C↑ |
| Zero-DCE | 14.86 | 0.54 | 7.77 | 4.01 | 1.15 | **18.06** | 0.58 | 8.06 | 3.92 | 1.20 |
| CLIP-LIT | 12.39 | 0.49 | 8.29 | 3.37 | **1.21** | 15.18 | 0.53 | 8.41 | 3.37 | **1.27** |
| RetinexDIP | 8.59 | 0.30 | 6.90 | 2.41 | 1.10 | 11.08 | 0.32 | 7.23 | 2.65 | 1.10 |
| NeuralBR | 11.36 | 0.44 | 7.52 | 3.68 | 1.17 | 14.04 | 0.47 | 7.56 | 3.67 | 1.22 |
| GDP | 13.93 | 0.63 | **6.17** | **5.34** | 0.67 | 13.04 | 0.55 | 7.59 | 4.29 | 0.40 |
| RetinexGDP | **15.66** | **0.66** | 6.26 | 5.26 | 0.85 | 16.51 | **0.69** | **6.92** | **4.97** | 0.96 |

Table 3: Ablation study on loss and text.

| Loss | N↓ | M↑ | C↑ |
|---|---|---|---|
| $\mathcal{L}_{recon}$ | **5.44** | 5.03 | **1.05** |
| $\mathcal{L}_{recon}$ w/ text | 6.47 | 4.81 | 0.69 |
| $\mathcal{L}_{recon} + \mathcal{L}_{per}$ | 5.58 | **5.07** | 0.96 |
| $\mathcal{L}_{recon} + \mathcal{L}_{per}$ w/ text | 5.63 | 5.01 | 0.89 |

**Patch-wise DDIM inversion.** The patch strategy in the DDIM inversion not only accommodates inputs of any size but also aids in preserving structure and textures, as demonstrated in Fig.9. Without the patch strategy, structures tend to be distorted (as seen in the red box), and artifacts tend to appear in darker areas (as observed in the white box).

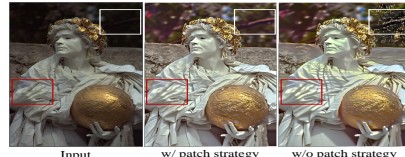

Figure 9: Ablation study on patch-wise DDIM inversion.

**Linear combination factor of corrected reflectance.** The loss function used in reverse sampling process, both the intermediate result $x_{0,t}$ and corrected reflectance are linear combination. We find the factor for linear combination $\eta$ play an important role in the denoising process, as shown in Fig.10. When $\eta = \sqrt{\bar{\alpha}_t}$, $\eta$ increases as the value of $\bar{\alpha}_t$ gradually increases, resulting in smoother enhanced results. In contrast, when $\eta = \sqrt{1 - \bar{\alpha}_t}$, the model generates detailed results.

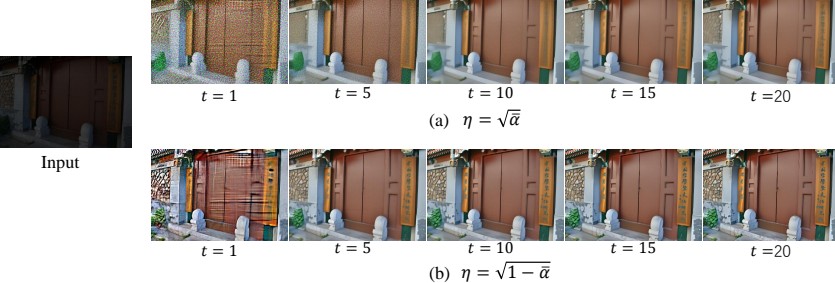

Figure 10: Ablation study on factor of combination $\eta$. Zoom in for a better view.

## 5 CONCLUSION

In this paper, we have introduced RetinexGDP, a zero-shot PLIE model that combines a generative pretrained diffusion model with domain knowledge from Retinex theory. RetinexGDP enables flexible enhancement guided by user preferences specified via text instructions, obviating the necessity for additional training data or external images. We incorporate TV optimization into a single Gaussian covolutional layer for zero-shot illumination estimation, streamlining the pipeline of Retinex decomposition. To maintain the content and structure consistency, we employ patch-wise DDIM inversion to find the initial noise vector of corrected reflectance and perform sampling conditioned on the correced reflectance. While our zero-shot, training-free PLIE method may not outperform state-of-the-art models across all datasets, it delivers competitive results. Its key strength lies in the ability to flexibly customize enhancement styles through text prompts, offering a unique and adaptable solution. This novel feature brings valuable flexibility to low-light enhancement, deserving further attention from the research community.

**Limitations:** RetinexGDP has limitation in real-time enhancement due to the inversion process. This limitation may be solved by inversion-free method Xu et al. (2024).

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

## A    APPENDIX

### A.1    PRELIMINARY: DIFFUSION MODEL

DDPMs are designed to reverse a parameterized Markovian image noise process. Initially, they operate on isotropic Gaussian noise samples, transforming them into samples drawn from a training distribution. This transformation is achieved through an iterative diffusion process that gradually eliminates the noise. Recent studies have demonstrated the capability of DDPMs to generate high-quality imagesDhariwal & Nichol (2021); Ho et al. (2020); Nichol & Dhariwal (2021). In the following sections, we provide a concise overview of DDPMs. For a more comprehensive understanding, readers are encouraged to refer to Ho et al. (2020); Nichol & Dhariwal (2021); Sohl-Dickstein et al. (2015).

Assuming a data distribution $x_0 \sim q(x_0)$, the inversion process generates a sequence of latent variables $x_1, \ldots, x_T$ by incrementally adding Gaussian noise with variance $\beta_t \in (0, 1)$ at time step $t$:

$$q(x_1, \ldots, x_T \mid x_0) = \prod_{t=1}^{T} q(x_t \mid x_{t-1})$$
$$q(x_t \mid x_{t-1}) = \mathcal{N}\left(\sqrt{1 - \beta_t} x_{t-1}, \beta_t \boldsymbol{I}\right) \tag{10}$$

An essential characteristic of the inversion is its ability to directly sample any step $x_t$ from $x_0$:

$$q(x_t \mid x_0) = \mathcal{N}\left(\sqrt{\bar{\alpha}_t} x_0, (1 - \bar{\alpha}_t) \boldsymbol{I}\right)$$
$$x_t = \sqrt{\bar{\alpha}_t} x_0 + \sqrt{1 - \bar{\alpha}_t} \epsilon \tag{11}$$

where $\epsilon \sim \mathcal{N}(0, \boldsymbol{I})$, $\alpha_t = 1 - \beta_t$ and $\bar{\alpha}_t = \prod_{s=0}^{t} \alpha_s$.

The reverse process is also Markovian, starting from a Gaussian noise sample $x_T \sim \mathcal{N}(0, \boldsymbol{I})$, and generating a reverse sequence by sampling the posteriors $q(x_{t-1} \mid x_t)$.

Since the exact form of $q(x_{t-1} \mid x_t)$ remains unknown, a deep neural network $P_\theta$ is trained to estimate the mean and covariance of $x_{t-1}$ given $x_t$:

$$P_\theta(x_{t-1} \mid x_t) = \mathcal{N}\left(\mu_\theta(x_t, t), \Sigma_\theta(x_t, t)\right). \tag{12}$$

Instead of directly predicting $\mu_\theta(x_t, t)$, Ho et al. (2020) propose predicting the noise $\epsilon_\theta(x_t, t)$ added to $x_0$ to obtain $x_t$:

$$\mu_\theta(x_t, t) = \frac{1}{\sqrt{\alpha_t}} \left(x_t - \frac{\beta_t}{\sqrt{1 - \bar{\alpha}_t}} \epsilon_\theta(x_t, t)\right) \tag{13}$$

The estimated image $\tilde{\boldsymbol{x}}_0$ is then derived from $\epsilon_\theta(x_t, t)$ using Equation (11):

$$\tilde{\boldsymbol{x}}_0 = \frac{\boldsymbol{x}_t}{\sqrt{\bar{\alpha}_t}} - \frac{\sqrt{1 - \bar{\alpha}_t} \epsilon_\theta(\boldsymbol{x}_t, t)}{\sqrt{\bar{\alpha}_t}}$$
$$q(\boldsymbol{x}_{t-1} \mid \boldsymbol{x}_t, \tilde{\boldsymbol{x}}_0) = \mathcal{N}\left(\boldsymbol{x}_{t-1}; \tilde{\boldsymbol{\mu}}_t(\boldsymbol{x}_t, \tilde{\boldsymbol{x}}_0), \tilde{\beta}_t \boldsymbol{I}\right)$$
$$\text{where} \quad \tilde{\boldsymbol{\mu}}_t(\boldsymbol{x}_t, \tilde{\boldsymbol{x}}_0) = \frac{\sqrt{\bar{\alpha}_{t-1}} \beta_t}{1 - \bar{\alpha}_t} \tilde{\boldsymbol{x}}_0 + \frac{\sqrt{\alpha_t}(1 - \bar{\alpha}_{t-1})}{1 - \bar{\alpha}_t} \boldsymbol{x}_t \tag{14}$$
$$\text{and} \quad \tilde{\beta}_t(\eta) = \frac{1 - \bar{\alpha}_{t-1}}{1 - \bar{\alpha}_t} \beta_t$$

where $\eta \in [0, 1]$, $\eta$ controls the interpolation ratio between DDPM and DDIM, , making the process deterministic when $\eta = 0$. For more details please see Ho et al. (2020).

## A.2 Gaussian TV layer

**Motivation.** It is well known that the convolution can be use to smooth the details in an image, while the TV optimization can be applied to preserve the image edges, as illustrated in Fig.11. In our illumination estimation, we expect the illumination to be piecewise smooth. Hence, we incorporate the edge-aware property of TV optimization into a single Gaussian convolutional layer, leading to zero-shot illumination estimation.

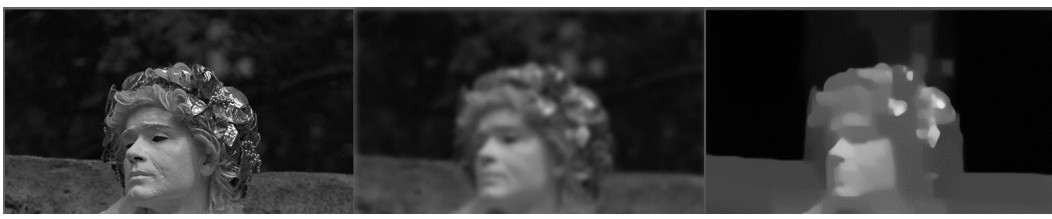

| Input | Convolution | TV optimization |

Figure 11: Illustration of the convolution process and TV optimization for image smoothing.

## A.3 Patch-wise DDIM Inversion

The patch-wise DDIM inversion process is outlined in Alg.1. The DDIM inversion takes correct reflectance $\boldsymbol{R}$ as input and produces the corresponding initial noise vector $\hat{\boldsymbol{R}}$. The corrected reflectance $\boldsymbol{R}$ is divided into $M$ overlapping patches. These patches are cropped with a stride of $p$, ensuring that some areas of the image are covered multiple times.

For each patch, a noised intermediate result is obtained. At each time step $t$, the diffusion model calculates the mean $\mu^m$ and variance $\Sigma^m$ of the Gaussian noise for each patch.

The mean and variance for each patch are iteratively updated to reflect the overall values for the entire image.

Since patches overlap, the areas covered multiple times must be tracked. A binary patch mask $\mathbf{P}^m$ identifies overlapping areas, and a weight vector $\mathbf{G}$ records how many times each pixel in these areas is included in a patch. The update follows: $\mathbf{G} = \mathbf{G} + \mathbf{P}^m$, where $m$ indicates the $m$-th patch.

The final mean $\Omega_t$ and variance $\phi_t$ for the entire image at each time step are computed by taking the weighted average, adjusting for overlapping areas:

$$\Omega_t = \Omega_t \oslash \mathbf{G}, \quad \phi_t = \phi_t \oslash \mathbf{G} \tag{15}$$

where $\oslash$ denotes element-wise division.

This weighted averaging ensures that the contributions from overlapping patches are aggregated, maintaining the consistency of the image's structure and texture throughout the diffusion process.

## A.4 Text guided sampling conditioned on reflectance

The procedure for text guided sampling conditioned on reflectance is outlined in Alg.2

**Initial Noise Vector.** The initial noise vector $\hat{\boldsymbol{R}}$, generated via DDIM inversion, is used as the starting latent vector in the reverse denoising step to maintain data consistency. Instead of pure Gaussian noise, this approach integrates the corrected reflectance $\boldsymbol{R}$ as a condition to ensure that the structure and texture of the original image are preserved while enhancing illumination.

**Conditional Sampling.** To improve the diffusion model's quality and reliability, the corrected reflectance $\boldsymbol{R}$ is used as a condition in the reverse denoising process. The conditional distribution $P_\theta(x_{t-1} \mid x_t, \boldsymbol{R})$ aims to guide the denoising towards high-quality outputs consistent with the original image.

---

**Algorithm 1** Patch-wise DDIM inversion for finding the initial noise vector

---

1: **Input:** Reflectance $R$, diffusion steps $T$, diffusion model $(\mu_\theta(x_t), \Sigma_\theta(x_t))$, dictionary of $M$ overlapping patch locations, and a binary patch mask $\mathbf{P}^m$
2: **Output:** initial noise vector $\hat{R}$
3: Sample $x_0$ from $\mathcal{N}(0, I)$
4: **for all** $t$ from 1 to $T$ **do**
5: $\quad \mu, \Sigma \leftarrow \mu_\theta(x_t), \Sigma_\theta(x_t)$
6: $\quad$ Mean vector $\Omega_t = 0$ and variance vector $\phi_t = 0$ and weight vector $\mathbf{G} = 0$
7: $\quad$ **for** $m = 1, ..., M$ **do**
8: $\quad\quad R^m = Crop(\mathbf{P}^m \odot R)$
9: $\quad\quad \tilde{x}_{t-1}^m = \sqrt{\alpha_{t-1}}(\frac{x_t^m}{\sqrt{\bar{\alpha}_t}} - \frac{\sqrt{1-\bar{\alpha}_t}\epsilon_\theta(x_t^m, t)}{\sqrt{\bar{\alpha}_t}}) + \sqrt{1 - \alpha_{t-1}}\epsilon_\theta(x_t^m, t)$
10: $\quad\quad \Omega_t = \Omega_t + \mathbf{P}^m \circ \mu^m$
11: $\quad\quad \phi_t = \phi_t + \mathbf{P}^m \circ \Sigma^m$
12: $\quad\quad \mathbf{G} = \mathbf{G} + \mathbf{P}^m$
13: $\quad$ **end for**
14: $\quad \Omega_t = \Omega_t \oslash \mathbf{G}$
15: $\quad \phi_t = \phi_t \oslash \mathbf{G}$
16: $\quad$ sample $x_{t-1}$ by $\mathcal{N}(\Omega_t, \phi_t)$
17: **end for**
18: **return** $\hat{R}$

---

**Heuristic Approximation.** The conditional probability $P(R \mid x_t)$ can be approximated as:

$$P(R \mid x_t) = \frac{1}{Z}\exp\left(-[\lambda_1 \mathcal{L}_c(x_t, R) + \lambda_2 \mathcal{L}_s(x_t, R, d)]\right) \tag{16}$$

where $Z$ is a normalizing factor, $\mathcal{L}_c$ and $\mathcal{L}_s$ are content and style loss metrics, $d$ is the text prompt for enhancement style, and $\lambda_1$ and $\lambda_2$ control the guidance strength. The gradients of both sides are computed as:

$$\log P(R \mid x_t) = -\log Z - \lambda_1 \mathcal{L}_c(x_t, R) - \lambda_2 \mathcal{L}_s(x_t, R, d)$$
$$\nabla_{x_t} \log P(R \mid x_t) = -\lambda_1 \nabla_{x_t} \mathcal{L}_c(x_t, R) - \lambda_2 \nabla_{x_t} \mathcal{L}_s(x_t, R, d) \tag{17}$$

**Mean Shift Adjustment.** The conditional transition $P_\theta(x_{t-1} \mid x_t, R)$ is derived from the unconditional transition by shifting the mean: $-(\lambda_1 \nabla_{x_t} \mathcal{L}_c(x_t, R) + \lambda_2 \nabla_{x_t} \mathcal{L}_s(x_t, R, d))$. Therefore, the mean of conditional transition $P_\theta(x_{t-1} \mid x_t, R)$ becomes:

$$\mu = \mu + \Sigma \nabla_{x_t} \log P(R \mid x_t) \tag{18}$$

In other words, by adjusting the scaling factors $\lambda_1$ and $\lambda_2$, and specifying the enhancement style through a text prompt, the level of guidance in the generation process can be controlled. To avoid regression-to-the-mean effects, gradients are computed for a linear combination of the intermediate denoised image and the initial reflectance:

$$\tilde{x}_{0,t-1} = \eta x_{0,t-1} + (1 - \eta)x_{0,t}, \quad \eta = \sqrt{1 - \bar{\alpha}_t} \tag{19}$$

In practice, the condition is a linear combination of $R$ and $\tilde{R}$, where $\tilde{R}_t = \sqrt{\bar{\alpha}_t}R_0 + \sqrt{1 - \bar{\alpha}_t}\epsilon$, and $R_t = \eta R_0 + (1 - \eta)\tilde{R}_t$.

A.5 ABLATION STUDY

A.5.1 ANALYSIS ON REVERSE DENOISING PROCESS

To further illustrate the effectiveness of the DDIM-DDIM diffusion model, we visualize the reverse sampling process in Fig.12. Notably, we observe that the results stabilize after the 5-th time step.

A.5.2 ANALYSIS ON CONTENT GUIDANCE AND STYLE GUIDANCE

The combination of "MSE+VGG+CLIP" enables our RetinexGDP to produce pleasing results, with improved quantitative evaluation. However, the absence of VGG in the loss function weakens the

---

**Algorithm 2** Text-based sampling conditioned on reflectance

---

1: **Input:** Noised reflectance $\hat{R}$, text description $d$, diffusion steps $T$, diffusion model $(\mu_\theta(x_t), \Sigma_\theta(x_t))$, content consistency coefficient $\lambda_1$, style coefficient $\lambda_2$, dictionary of $M$ overlapping patch locations, and a binary patch mask $\mathbf{P}^m$

2: **Output:** $x_0$ enhanced image

3: Sample $x_T$ from $\mathcal{N}\left(\sqrt{\bar{\alpha}_k}\hat{R}, (1-\bar{\alpha}_k)\mathbf{I}\right)$

4: **for all** $t$ from $T$ to 1 **do**

5: $\quad \mu, \Sigma \leftarrow \mu_\theta(x_t), \Sigma_\theta(x_t)$

6: $\quad$ Mean vector $\Omega_t = 0$ and variance vector $\phi_t = 0$ and weight vector $\mathbf{G} = 0$

7: $\quad$ **for** $m = 1, ..., M$ **do**

8: $\quad\quad x_t^m = Crop(\mathbf{P}^m \odot x_t)$

9: $\quad\quad \mathbf{R}^m = Crop(\mathbf{P}^m \odot \mathbf{R})$

10: $\quad\quad \tilde{x}_{0,t}^m = \frac{x_t^m}{\sqrt{\bar{\alpha}_t}} - \frac{\sqrt{1-\bar{\alpha}_t}\epsilon_\theta(x_t^m, t)}{\sqrt{\bar{\alpha}_t}}$

11: $\quad\quad \tilde{x}_{0,t}^m = \sqrt{\alpha_t}\left(\frac{x_{t-1}^m}{\sqrt{\bar{\alpha}_{t-1}}} - \frac{\sqrt{1-\bar{\alpha}_{t-1}}\epsilon_\theta\left(x_{t-1}^m, t\right)}{\sqrt{\bar{\alpha}_{t-1}}}\right) + \sqrt{1-\alpha_t}\epsilon_\theta\left(x_{t-1}^m, t-1\right)$

12: $\quad\quad \tilde{x}_{0,t}^m = \eta\tilde{x}_{0,t-1}^m + (1-\eta)\tilde{x}_{0,t}^m$

13: $\quad\quad \mathbf{R}_t^m \sim \mathcal{N}\left(\sqrt{\bar{\alpha}_t}\mathbf{R}^m, (1-\bar{\alpha}_t)\mathbf{I}\right)$

14: $\quad\quad \mathbf{R}_t^m = \eta\mathbf{R}^m + (1-\eta)\mathbf{R}_t^m$

15: $\quad\quad \mathcal{L}^{total} = \lambda_1\mathcal{L}_c\left(\tilde{x}_{0,t}, \mathbf{R}_t^m\right) + \lambda_2\mathcal{L}_{clip}\left(\tilde{x}_{0,t}, \mathbf{R}_t^m, d\right)$

16: $\quad\quad \mu^m = \mu^m + \Sigma\nabla_{x_t}\mathcal{L}^{total}$

17: $\quad\quad \Omega_t = \Omega_t + \mathbf{P}^m \circ \mu^m$

18: $\quad\quad \phi_t = \phi_t + \mathbf{P}^m \circ \Sigma^m$

19: $\quad\quad \mathbf{G} = \mathbf{G} + \mathbf{P}^m$

20: $\quad$ **end for**

21: $\quad \Omega_t = \Omega_t \oslash \mathbf{G}$

22: $\quad \phi_t = \phi_t \oslash \mathbf{G}$

23: $\quad$ sample $x_{t-1}$ by $\mathcal{N}(\Omega_t, \phi_t)$

24: **end for**

25: **return** $x_0$

---

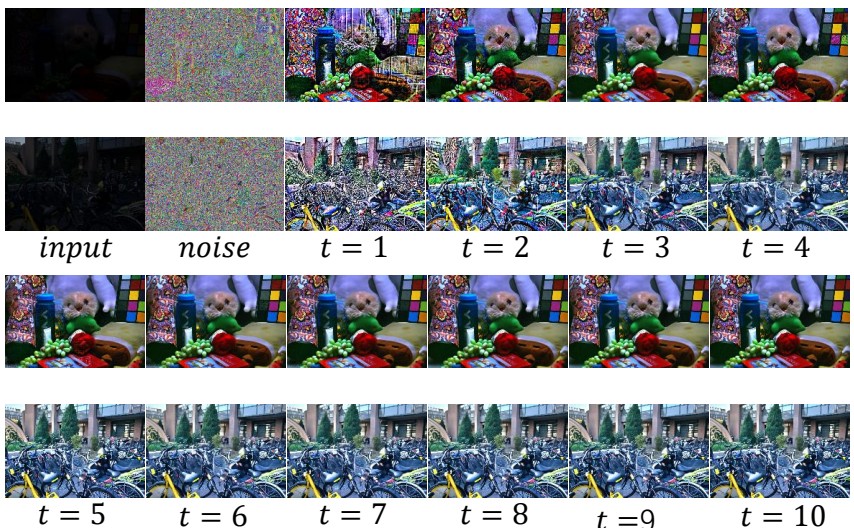

Figure 12: Visualization of reverse denoising process. Zoom in for a better view.

model's ability to suppress noise, as evidenced by the visual results in Fig.13. One possible reason for this improvement could be that the perceptual reconstruction provided by the VGG loss helps alleviate the domain shift problem of the CLIP model. Therefore, when using CLIP loss for text-guided enhancement, it is advisable to include VGG loss in the loss function and finetune the balance parameter of directional CLIP loss in Equation (9) to mitigate the occurrence of artifacts. Due to

domain shift problem of the CLIP model, when the weight of directional CLIP loss is set a large value, the content and structure of enhanced image may become inconsistent with the original image, as shown in Fig.14.

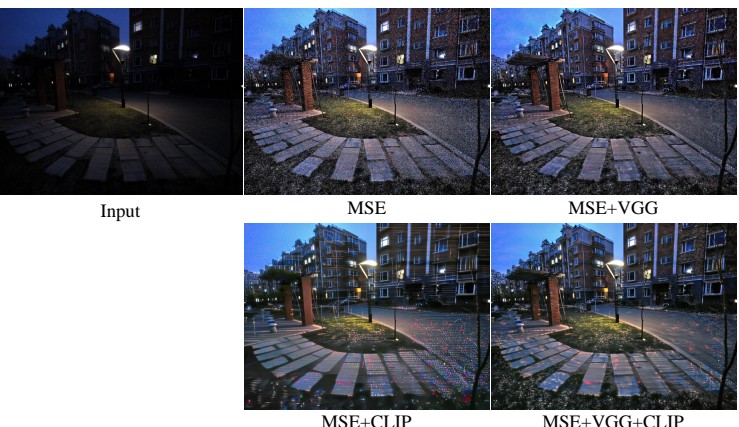

Input       MSE       MSE+VGG

MSE+CLIP       MSE+VGG+CLIP

Figure 13: Visual comparison while using different loss functions.

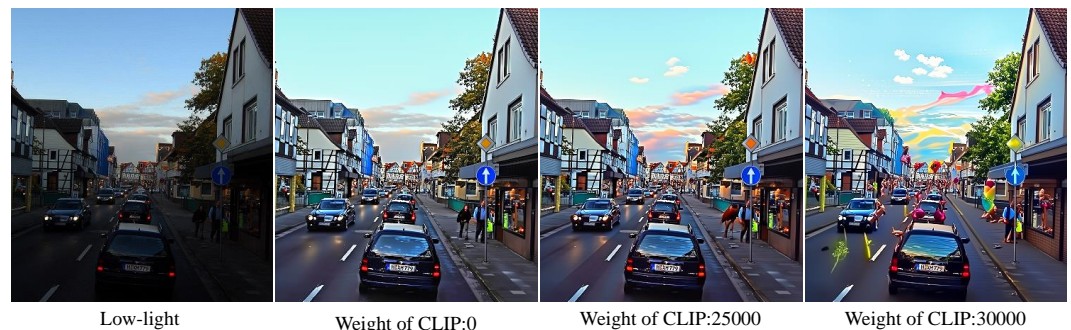

Low-light    Weight of CLIP:0    Weight of CLIP:25000    Weight of CLIP:30000

Figure 14: Visual results while using different weight for CLIP loss function.

### A.5.3 ANALYSIS ON DIFFUSION MODEL

We explored the impact of the diffusion model on personalized enhancement, as depicted in Fig.15. The DDPM-DDPM diffusion model fails to maintain content consistency, resulting in images that lose much of their original content and exhibit excessive smoothness compared to the DDIM-DDIM model. This is reasonable. First, the denoising process of DDIM model is deterministic, facilitating data fidelity. Second, DDIM inversion produces initial noise vector of the corrected reflectance, rather than pure noise, benefiting preserving structures and content.

### A.6 ADDITIONAL RESULTS

**Personalized low-light image enhancement.** We provide additional personalized enhanced results, as shown in Fig.16 and Fig.17. From Fig.16, without text instruction for specifying the enhanced style, the quality of the input image is improved and more details are appeared. With text instruction, the input image can be enhanced aligning with the specified styles, as shown in Fig.17. The most important is that, the content and structure in the enhanced images are preserved well.

**General low-light image enhancement.** We also provide additional general enhanced results. The visual comparisons with the state-of-the-art LLIE methods are shown in Fig.19, Fig.20, Fig.21 and Fig.22. Figure 19 shows that, our RetinexGDP is robust while dealing with non-uniform illumination scene, without over-enhancing the bright areas while enlightening the dark areas. Figure

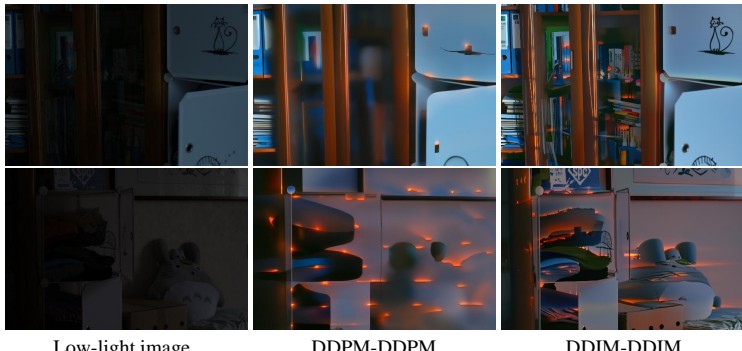

| Low-light image | DDPM-DDPM | DDIM-DDIM |

Figure 15: Impact of different diffusion models (Results obtained guided by "Summer sunset"). DDPM-DDPM indicates that both the inversion and reverse sampling processes adopt the DDPM model, and similarly, DDIM-DDIM.

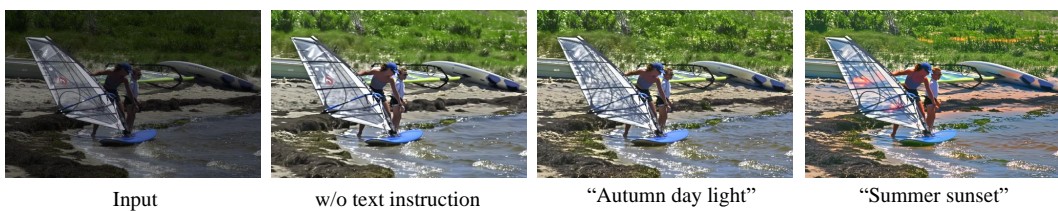

| Input | w/o text instruction | "Autumn day light" | "Summer sunset" |

Figure 16: Personalized enhanced results on Nasa dataset.

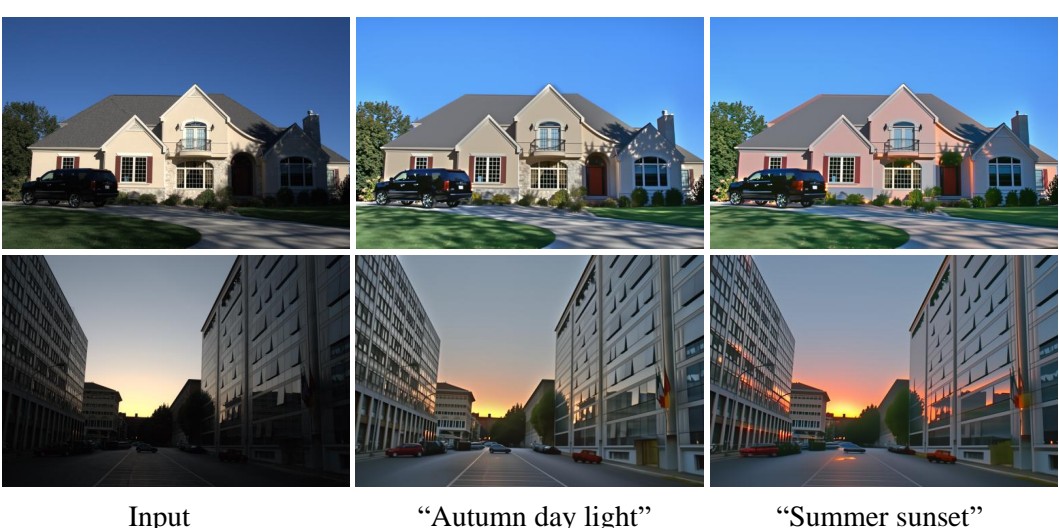

| Input | "Autumn day light" | "Summer sunset" |

Figure 17: Personalized enhanced results on MIT-Adobe FiveK dataset.

20 demonstrates our RetinexGDP's ability of noise suppression. The visual evaluations on paired dataset LOL and VELOL demonstrate that our RetinexGDP produces enhanced image with higher contrast, as shown in Fig.21 and Fig.22. Fig.18 demonstrates that our RetinexGDP can produce results with high contrast.

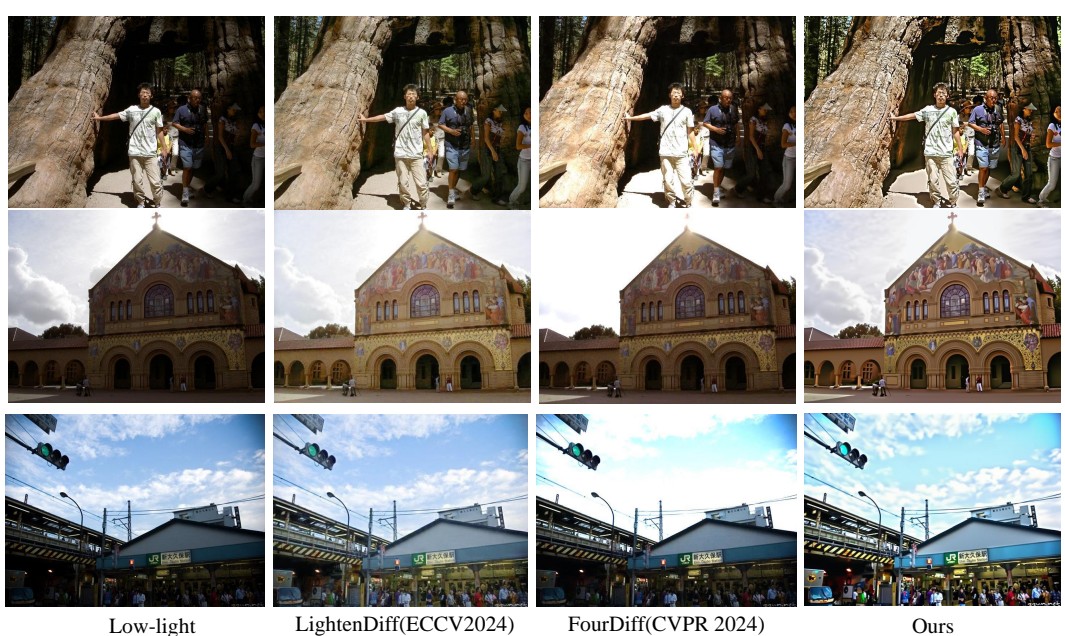

Figure 18: Visual comparison with the state-of-the-art LLIE methods on DICM dataset.

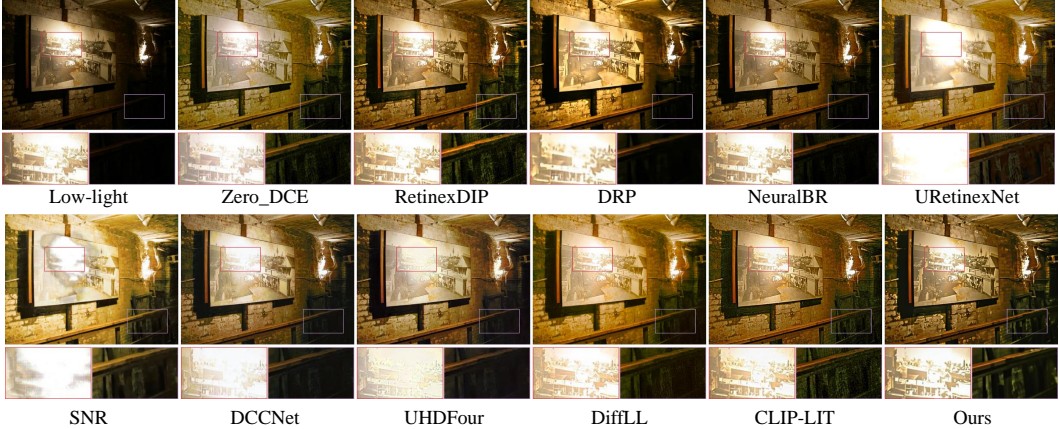

Figure 19: Visual comparison with the state-of-the-art LLIE methods on LIME dataset.

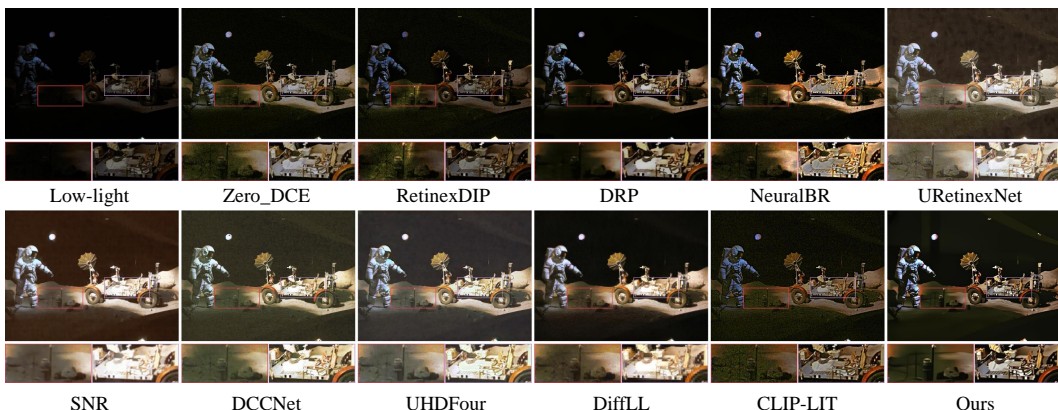

Figure 20: Visual comparison with the state-of-the-art LLIE methods on LIME dataset.

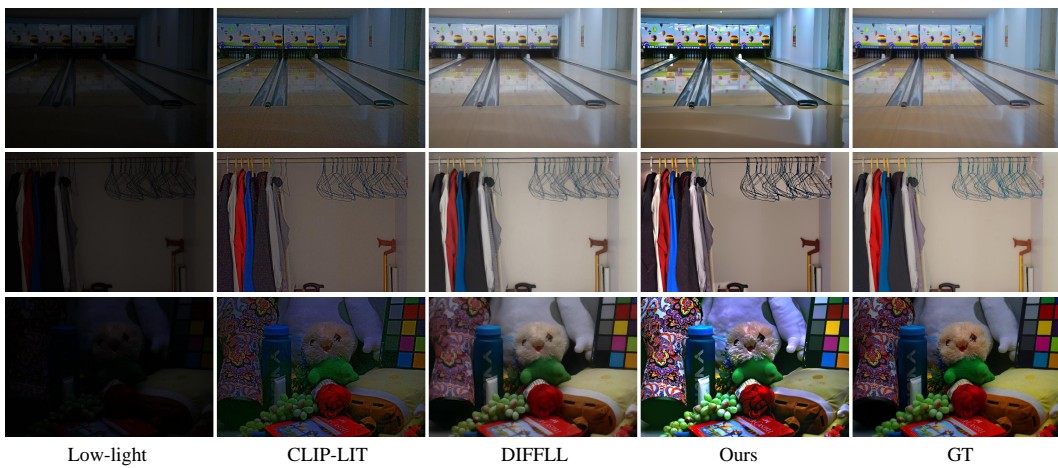

Figure 21: Visual comparison with the state-of-the-art LLIE methods on LOL dataset.

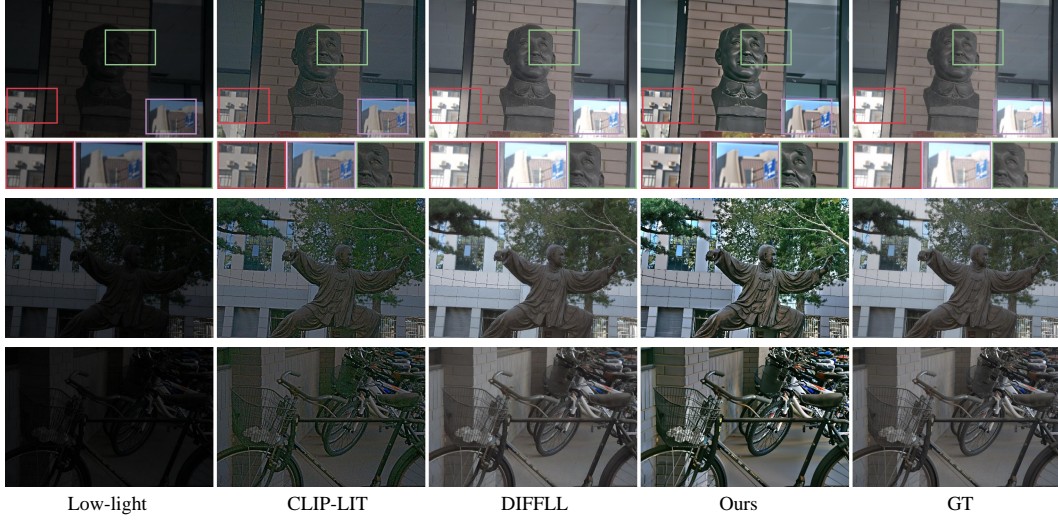

Figure 22: Image contrast visual comparison with the state-of-the-art LLIE methods on VELOL dataset.

