# OpenReview forum: "Zero-shot Text-based Personalized Low-Light Image Enhancement with Reflectance Guidance"
_ICLR.cc/2025/Conference — ICLR 2025 Conference Withdrawn Submission_

### Official Review · Reviewer_tPy1 · 2024-10-31

**Soundness:** 3
**Presentation:** 3
**Contribution:** 3
**Rating:** 6
**Confidence:** 4

**Summary:**

This paper presents RetinexGDP, a training-free, zero-shot personalized low-light image enhancement model that integrates Retinex domain knowledge into a pre-trained diffusion model. Key contributions include using total variation optimization for zero-shot Retinex decomposition, incorporating the CLIP model to guide enhancements based on user text instructions, and employing patch-wise DDIM inversion for consistent results. RetinexGDP supports any image size and delivers noise-suppressed results without extra noise constraints, achieving performance comparable to state-of-the-art models across nine low-light image datasets.

**Strengths:**

1. Well-designed structure, CLIP is introduced into the diffusion model, and the enhancement-related reflectivity is used as conditional information to assist in the generation of enhanced images.
2. The author incorporate the edge-aware property of total variation optimization into a single Gaussian convolutional layer, aiming to achieve zero-shot Retinex decomposition.
3. A large number of experiments have proved the effectiveness of the method.

**Weaknesses:**

1. The inconsistent use of metrics confuses me. This work uses different metrics on different datasets. Why is NIQE used in the ablation experiment but not in the comparison experiment? The cited NIQMC can also be used on the LOL and VELOL datasets.
2. In addition to this work, there are other work[1] that adopt similar core ideas. Compared with their methods, what are your significant advantages?

[1] He C, Fang C, Zhang Y, et al. Reti-diff: Illumination degradation image restoration with retinex-based latent diffusion model[J]. arXiv preprint arXiv:2311.11638, 2023.

3. Some experimental details need to be stated in the paper, such as the specific yperparameters of for each loss function term.

**Questions:**

please refers to the weakness

---

> ### Author Response · Authors · 2024-11-21
>
> We thank you for your valuable feedback. The points raised have help us to greatly improve the manuscript. Let us address the different points that you have raised. Please find the updated manuscript, with all the modifications highlighted in blue.
>
> >**Why is NIQE used in the ablation experiment but not in the comparison experiment?
>
> The reason that we did not use NIQE in the comparative experiments is that we had to add another table to show these results, which caused our manuscript to exceed the page limit.
>
> In the revised version, we have added the experimental results using NIQE for all datasets, as shown in Table 1.
>
> | Method             | DICM N↓ | DICM M↑ | DICM C↑ | ExDark N↓ | ExDark M↑ | ExDark C↑ | Fusion N↓ | Fusion M↑ | Fusion C↑ | LIME N↓ | LIME M↑ | LIME C↑ | Nasa N↓ | Nasa M↑ | Nasa C↑ | NPEA N↓ | NPEA M↑ | NPEA C↑ | VV N↓ | VV M↑ | VV C↑ |
> |--------------------|---------|---------|---------|-----------|-----------|-----------|-----------|-----------|-----------|---------|---------|---------|---------|---------|---------|---------|---------|---------|--------|--------|--------|
> | URetinexNet        | 3.50    | 5.17    | 0.72    | 3.74      | 5.18      | 0.87      | 3.90      | 5.07      | 0.85      | 4.33    | 5.02    | 0.90    | 3.27    | 5.14    | 0.84    | 4.08    | 5.00    | 0.85    | 3.03   | 5.14   | 0.79   |
> | SNR                | 4.16    | 5.17    | 0.63    | 4.29      | 5.29      | 0.68      | 4.93      | 5.00      | 0.64      | 5.69    | 5.23    | 0.72    | 5.23    | 5.28    | 0.57    | 4.15    | 4.98    | 0.67    | 8.77   | 4.98   | 0.45   |
> | DCCNet             | 3.29    | 5.02    | 0.75    | 3.75      | 5.07      | 0.74      | 4.29      | 4.80      | 0.77      | 4.26    | 5.04    | 0.88    | 3.18    | 5.16    | 0.87    | 3.50    | 4.74    | 0.75    | 3.56   | 5.11   | 0.76   |
> | UHDFour            | 3.46    | 5.09    | 0.75    | 3.90      | 5.00      | 0.71      | 4.38      | 4.85      | 0.72      | 4.55    | 4.97    | 0.85    | 3.33    | 5.39    | 0.85    | 3.62    | 4.94    | 0.74    | -      | -      | -      |
> | DiffusionLL        | 2.93    | 5.22    | 0.77    | 3.27      | 5.04      | 0.79      | 3.30      | 5.19      | 0.80      | 3.58    | 4.92    | 0.95    | 2.81    | 5.33    | 0.82    | 3.24    | 5.00    | 0.81    | 2.92   | 5.36   | 0.88   |
> | CLIPLIT            | 3.01    | 5.05    | 0.84    | 3.63      | 4.84      | 1.04      | 3.74      | 5.09      | 1.00      | 3.99    | 5.09    | 1.00    | 3.16    | 5.19    | 1.04    | 3.71    | 4.97    | 0.98    | 3.02   | 5.20   | 1.00   |
> | Zero_DCE           | 2.83    | 5.12    | 0.82    | 3.54      | 4.96      | 0.97      | 3.58      | 5.21      | 0.91      | 3.76    | 4.84    | 1.06    | 3.57    | 5.09    | 0.87    | 2.97    | 4.89    | 0.92    | 3.21   | 5.40   | 0.89   |
> | LightenDiffusion   | 3.39    | 5.23    | 0.90    | 3.34      | 5.14      | 0.80      | 3.43      | 5.21      | 0.78      | 4.04    | 5.10    | 0.94    | 3.08    | 5.48    | 0.89    | 3.03    | 5.14    | 0.79    | 3.58   | 5.38   | 0.77   |
> | FourierDiff        | 3.97    | 4.94    | 0.86    | 3.80      | 5.17      | 0.82      | 4.33      | 4.66      | 0.83      | 4.21    | 5.16    | 0.97    | 3.37    | 4.75    | 0.85    | 3.72    | 5.06    | 0.84    | 3.36   | 4.75   | 0.85   |
> | RetinexDIP     | 3.37    | 5.13    | 0.86    | 3.74      | 4.86      | 1.13      | 3.40      | 5.33      | 1.05      | 3.82    | 4.88    | 1.16    | 3.58    | 5.41    | 1.02    | 3.01    | 5.15    | 1.04    | 2.48   | 5.45   | 1.06   |
> | DRP            | 4.68    | 5.24    | -       | 4.79      | 5.17      | -         | 5.71      | 5.28      | -         | 5.99    | 5.21    | -       | 4.30    | 5.62    | -       | 5.29    | 5.37    | -       | 8.80   | 5.45   | -      |
> | NeuralBR       | 3.39    | 5.29    | 0.88    | 3.79      | 4.72      | 1.08      | 3.41      | 5.23      | 1.05      | 3.74    | 5.03    | 1.14    | 2.97    | 5.19    | 1.04    | 3.72    | 5.15    | 1.05    | 3.21   | 5.40   | 1.05   |
> | RetinexGDP     | 4.02    | 5.12    | 0.84    | 4.80      | 4.97      | 0.81      | 5.22      | 5.27      | 0.86      | 5.54    | 5.06    | 0.94    | 4.11    | 5.48    | 0.91    | 4.21    | 5.38    | 0.75    | 4.10   | 5.26   | 0.74   |

---

> ### Author Response · Authors · 2024-11-21
>
> >**The cited NIQMC can also be used on the LOL and VELOL datasets.**
>
> According to the suggestion of the reviewer, we also used NIQE, NIQMC and CPCQI to test the LOL and VELOL datasets and include these results in the comparison experiments, as shown in Table 2.
>
> | Method          | LOL P$\uparrow$ | LOL S$\uparrow$ | LOL N$\downarrow$ | LOL M$\uparrow$ | LOL C$\uparrow$ | VELOL P$\uparrow$ | VELOL S$\uparrow$ | VELOL N$\downarrow$ | VELOL M$\uparrow$ | VELOL C$\uparrow$ |
> |-----------------|-----------------|-----------------|-------------------|-----------------|-----------------|-------------------|-------------------|---------------------|-------------------|-------------------|
> | **Zero-DCE**    | _14.86_     | 0.54        | 7.77          | 4.01        | 1.15        | **18.06**   | _0.58_      | 8.06          | 3.92        | 1.20        |
> | **CLIP-LIT**    | 12.39       | 0.49        | 8.29          | 3.37        | **1.21**    | 15.18       | 0.53        | 8.41          | 3.37        | **1.27**    |
> | **RetinexDIP**  | 8.59        | 0.30        | 6.90          | 2.41        | 1.10        | 11.08       | 0.32        | _7.23_        | 2.65        | 1.10        |
> | **NeuralBR**    | 11.36       | 0.44        | 7.52          | 3.68        | _1.17_      | 14.04       | 0.47        | 7.56          | 3.67        | _1.22_      |
> | **GDP**         | 13.93       | _0.63_      | **6.17**      | **5.34**    | 0.67        | 13.04       | 0.55        | 7.59          | _4.29_      | 0.40        |
> | **RetinexGDP**  | **15.66**   | **0.66**    | _6.26_        | _5.26_      | 0.85        | _16.51_     | **0.69**    | **6.92**      | **4.97**    | 0.96        |

---

> ### Author Response · Authors · 2024-11-21
>
> >**..., there are other work[1] .... Compared with their methods, what are your significant advantages?**
>
> We appreciate the reviewer's reference to **Reti-diff**. The key differences and advantages of RetinexGDP are as follow:
>
>  - **Text-based Personalization**: Unlike **Reti-diff**, which can only perform general low-light enhancement, our approach can not only perform general low-light enhancement but achieve personalization through user text instructions, offering greater flexibility and adaptability across different user needs.
>
>  - **Zero-shot Retinex decomposition**: While **Reti-diff** also uses Retinex theory, the Retinex decomposition is performed using complex Transformer that trained on specific datasets. We incorporate total variation optimization into a single Gaussian convolutional layer, enabling zero-shot Retinex decomposition and overcoming the challenge of dataset..
>
>  [1] He C, Fang C, Zhang Y, et al. Reti-diff: Illumination degradation image restoration with retinex-based latent diëusion model[J]. arXiv preprint arXiv:2311.11638, 2023
>
>
> >**Some experimental details need to be stated in the paper, such as the specific hyperparameters of for each loss function term.**
>
> We have moved the experimental setups from the appendix to the main manuscript, highlighted in blue in Section 4.
>
> - The Gamma factor ($\gamma$) in Equation (4) is set to 0.5. This value is derived from the default gamma correction value for most display devices, typically 2.2, which balances visual perception and device compatibility. To adjust the illumination, we set $\gamma = 1/2.2$, approximated to $\gamma = 1/2 = 0.5$ for simplicity.
>
> - The parameter $\lambda$ in Equation (3) is set to 30, while $\sigma$ in Equation (3) is empirically set to 0.5.

---

### Official Review · Reviewer_Rc4q · 2024-10-31

**Soundness:** 2
**Presentation:** 3
**Contribution:** 2
**Rating:** 5
**Confidence:** 5

**Summary:**

This paper introduces RetinexGDP, a training-free, zero-shot personalized low-light image enhancement model that integrates Retinex theory into a pre-trained diffusion model. The authors overcome limitations of existing methods, such as per-image optimization and lack of personalization. Extensive experiments demonstrate that RetinexGDP achieves state-of-the-art performance across nine low-light datasets.

**Strengths:**

1. The paper proposes a training-free zero-shot personalized low-light image enhancement model that allows for style personalization based on user-provided text instructions.
2. The introduction of the CLIP model into the reflectance-conditioned sampling process of DDIM allows for enhancement guided by user text instructions, increasing model flexibility.

**Weaknesses:**

1. Why use Reflectance directly for enhancement, rather than enhancing the entire image? Wouldn't this approach lose information contained in the Illumination?
2. The proposed method is training-free, so how can the loss function be optimized?
3. The colors used in the caption about the loss function in Figure 1 do not match the colors used in the figure itself.
4. There is a lack of specific explanation regarding the image content and style distance metrics L_c and L_s in Equation 5.
5. The authors should compare some recent unsupervised SOTA methods to demonstrate the effectiveness of the proposed method, such as LightenDiffusion (ECCV2024) and FourierDiff (cvpr2024).

**Questions:**

Please refer to the weaknesses section.

---

> ### Author Response · Authors · 2024-11-21
>
> We greatly appreciate the reviewer's thoughtful and detailed feedback. Below, we provide responses to the specific concerns raised in the review.
>
> >**The introduction of the CLIP model into the reflectance-conditioned sampling process of DDIM allows for enhancement guided by user text instructions, increasing model flexibility.**
>
> Thanks for your positive comment. Indeed, our RetinexGDP is flexible since it allows users to specify their preferred style of enhancement using text prompt.
>
>
> > **Why use Reflectance directly for enhancement, rather than enhancing the entire image? Wouldn't this approach lose information contained in the Illumination?**
>
>
> We would like to clarify that the reflectance itself is not directly utilized for subsequent personalized enhancement. Instead, we use the **corrected reflectance**, computed by  $R = S \oslash I^\gamma$, for this purpose. We consider the corrected reflectance as the initial enhanced image. We have added the example of the corrected reflectance in Figure 5. We can see that the corrected reflectance can be considered as a initial enhanced image, without lossing information.
>
>
> > **The proposed method is training-free, so how can the loss function be optimized?**
>
>
> Our model is **training-free**, as it does not require additional training to adapt to new tasks. Instead, the pretrained DDIM model is used directly for inference without updating its parameters. The loss functions are not used for training but to guide the sampling process, ensuring the final enhanced image adheres to style constraints and content consistency.
>
> During the sampling process, the loss functions adjust the output by **adding gradients to the denoised image at each time step**. These gradients do not update model parameters but shift the mean of the unconditional distribution. This shift ensures the generated images during DDIM sampling align closely with the distribution of personalized augmented images while preserving content consistency. We encourage the reviewer to refer to Ref. [1] for further details.
>
> [1]Dhariwal, P., & Nichol, A. (2021). Diffusion models beat gans on image synthesis. Advances in neural information processing systems, 34, 8780-8794.
>
>
> > **The colors used in the caption about the loss function in Figure 1 do not match the colors used in the figure itself.**
>
> We apologize for the lack of clarity in Figure 1. In the original version, we originally wanted to use the gray line to represent the loss functions, including $L_{recon}$, $L_{per}$, and $L_{clip}$, but we used the wrong color, we have revised this mistake. We have updated Figure 1 and the caption to clarify these points and improve the visibility of the line to ensure better understanding. Please check the revised version of our manuscript.
>
> > **There is a lack of specific explanation regarding the image content and style distance metrics L_c and L_s in Equation 5.**
>
> We appreciate the reviewer's observation about the lack of clarity regarding the image content and style distance metrics, $L_c$ and $L_s$.
> - $L_c$ is consist of reconstruction loss $L_{recon}$ and perceptual loss $L_{per}$. It ensures that the fine details and structures in the reflectance map are preserved.
> - We use pretrained CLIP model to style distance $L_s$. $L_s$ ensures the generated image matches the desired style, as guided by the user's text prompt.

---

> ### Author Response · Authors · 2024-11-21
>
> > **... compare some recent unsupervised SOTA methods ..., such as LightenDiffusion (ECCV2024) and FourierDiff (cvpr2024)**
>
> Thanks for your comment, we have addded the comparions with **LightenDiffusion** and **FourierDiff** in terms of both quantitative metrics (Table 1) and qualitative visual results (Figure 18 in Appendix A.6).
>
>
> | Method             | DICM N↓ | DICM M↑ | DICM C↑ | ExDark N↓ | ExDark M↑ | ExDark C↑ | Fusion N↓ | Fusion M↑ | Fusion C↑ | LIME N↓ | LIME M↑ | LIME C↑ | Nasa N↓ | Nasa M↑ | Nasa C↑ | NPEA N↓ | NPEA M↑ | NPEA C↑ | VV N↓ | VV M↑ | VV C↑ |
> |--------------------|---------|---------|---------|-----------|-----------|-----------|-----------|-----------|-----------|---------|---------|---------|---------|---------|---------|---------|---------|---------|--------|--------|--------|
> | URetinexNet        | 3.50    | 5.17    | 0.72    | 3.74      | 5.18      | 0.87      | 3.90      | 5.07      | 0.85      | 4.33    | 5.02    | 0.90    | 3.27    | 5.14    | 0.84    | 4.08    | 5.00    | 0.85    | 3.03   | 5.14   | 0.79   |
> | SNR                | 4.16    | 5.17    | 0.63    | 4.29      | 5.29      | 0.68      | 4.93      | 5.00      | 0.64      | 5.69    | 5.23    | 0.72    | 5.23    | 5.28    | 0.57    | 4.15    | 4.98    | 0.67    | 8.77   | 4.98   | 0.45   |
> | DCCNet             | 3.29    | 5.02    | 0.75    | 3.75      | 5.07      | 0.74      | 4.29      | 4.80      | 0.77      | 4.26    | 5.04    | 0.88    | 3.18    | 5.16    | 0.87    | 3.50    | 4.74    | 0.75    | 3.56   | 5.11   | 0.76   |
> | UHDFour            | 3.46    | 5.09    | 0.75    | 3.90      | 5.00      | 0.71      | 4.38      | 4.85      | 0.72      | 4.55    | 4.97    | 0.85    | 3.33    | 5.39    | 0.85    | 3.62    | 4.94    | 0.74    | -      | -      | -      |
> | DiffusionLL        | 2.93    | 5.22    | 0.77    | 3.27      | 5.04      | 0.79      | 3.30      | 5.19      | 0.80      | 3.58    | 4.92    | 0.95    | 2.81    | 5.33    | 0.82    | 3.24    | 5.00    | 0.81    | 2.92   | 5.36   | 0.88   |
> | CLIPLIT            | 3.01    | 5.05    | 0.84    | 3.63      | 4.84      | 1.04      | 3.74      | 5.09      | 1.00      | 3.99    | 5.09    | 1.00    | 3.16    | 5.19    | 1.04    | 3.71    | 4.97    | 0.98    | 3.02   | 5.20   | 1.00   |
> | Zero_DCE           | 2.83    | 5.12    | 0.82    | 3.54      | 4.96      | 0.97      | 3.58      | 5.21      | 0.91      | 3.76    | 4.84    | 1.06    | 3.57    | 5.09    | 0.87    | 2.97    | 4.89    | 0.92    | 3.21   | 5.40   | 0.89   |
> | LightenDiffusion   | 3.39    | 5.23    | 0.90    | 3.34      | 5.14      | 0.80      | 3.43      | 5.21      | 0.78      | 4.04    | 5.10    | 0.94    | 3.08    | 5.48    | 0.89    | 3.03    | 5.14    | 0.79    | 3.58   | 5.38   | 0.77   |
> | FourierDiff        | 3.97    | 4.94    | 0.86    | 3.80      | 5.17      | 0.82      | 4.33      | 4.66      | 0.83      | 4.21    | 5.16    | 0.97    | 3.37    | 4.75    | 0.85    | 3.72    | 5.06    | 0.84    | 3.36   | 4.75   | 0.85   |
> | RetinexDIP     | 3.37    | 5.13    | 0.86    | 3.74      | 4.86      | 1.13      | 3.40      | 5.33      | 1.05      | 3.82    | 4.88    | 1.16    | 3.58    | 5.41    | 1.02    | 3.01    | 5.15    | 1.04    | 2.48   | 5.45   | 1.06   |
> | DRP            | 4.68    | 5.24    | -       | 4.79      | 5.17      | -         | 5.71      | 5.28      | -         | 5.99    | 5.21    | -       | 4.30    | 5.62    | -       | 5.29    | 5.37    | -       | 8.80   | 5.45   | -      |
> | NeuralBR       | 3.39    | 5.29    | 0.88    | 3.79      | 4.72      | 1.08      | 3.41      | 5.23      | 1.05      | 3.74    | 5.03    | 1.14    | 2.97    | 5.19    | 1.04    | 3.72    | 5.15    | 1.05    | 3.21   | 5.40   | 1.05   |
> | RetinexGDP     | 4.02    | 5.12    | 0.84    | 4.80      | 4.97      | 0.81      | 5.22      | 5.27      | 0.86      | 5.54    | 5.06    | 0.94    | 4.11    | 5.48    | 0.91    | 4.21    | 5.38    | 0.75    | 4.10   | 5.26   | 0.74   |

---

### Official Review · Reviewer_2CJL · 2024-11-04

**Soundness:** 2
**Presentation:** 3
**Contribution:** 2
**Rating:** 5
**Confidence:** 5

**Summary:**

This paper introduces RetinexGDP, a training-free method for personalized low-light image enhancement, based on Retinex theory and a pre-trained text-to-image diffusion model. The method begins by employing a single Gaussian convolutional layer to perform Retinex decomposition. Subsequently, the decomposed reflectance is used as a condition in the diffusion model. RetinexGDP has been evaluated across several low-light image datasets and demonstrates performance comparable to state-of-the-art (SOTA) models.

**Strengths:**

1. Integrating Retinex information into deep learning models enhances the interpretability of the proposed method.

2. RetinexGDP is a training-free, zero-shot method that generates enhanced images based on user preferences via text instructions.

**Weaknesses:**

1. In Figure 1, the gray line representing the loss function is not visible. What do the blue and yellow lines represent? It seems there may be some confusion regarding these lines.

2. Decomposing illumination and reflectance from low-light images is generally challenging. Many supervised methods struggle with accurate decomposition. How can the use of the Gaussian filter and the TV proximity operator ensure effective decomposition in this method?

3. Why is gamma correction applied after estimating illumination, and why is the reflectance obtained only after applying gamma correction? Additionally, the values for the gamma correction factor in Equation (4) and the \lambda in Equation (3) are not provided. I hope the authors could clarify these  parameters.

4. Does the illumination shown in Figure 3 represent the illumination before or after the gamma correction? Additional visualizations would aid in better evaluation. In my view, the reflectance appears over-saturated, as do the visual results presented later. I am unsure whether this over-saturation is an effect of integrating gamma correction.

5. Although the paper claims that the proposed method is a training-free, zero-shot method, several loss functions are introduced. It is unclear which components are fixed and which are learnable. I hope authors provide a diagram of the gradient flow for clarity.

6. The proposed method does not demonstrate enhanced performance compared to RetinexDIP, which is also a training-free method. Furthermore, the differences between results with different text instructions appear minimal, as illustrated by the bottom line in Figure 5.

**Questions:**

The questions are listed up in the Weaknesses. A point-by-point response would be helpful. I will consider raising my rating if all concerns are well-addressed.

---

> ### Author Response · Authors · 2024-11-21
>
> We sincerely appreciate the reviewer's thorough and insightful feedback. We have carefully addressed each of the concerns raised and provide clarifications below.
>
> >**Integrating Retinex information into deep learning models enhances the interpretability of the proposed method.**
>
> Thank you for emphasizing the interpretability benefit of integrating Retinex domain knowledge into our model.
>
>
> >**In Figure 1, the gray line representing the loss function is not visible. What do the blue and yellow lines represent?**
>
> We apologize for the lack of clarity in Figure 1. In the original version, we intended to use the gray line to represent the loss functions, including $L_{recon}$, $L_{per}$, and $L_{clip}$, but used the wrong color. We have corrected this in the revised version. The blue and yellow lines were meant to indicate the input to the CLIP encoders (text encoder and image encoder), and we now use a dashed black line to represent them. We have updated both the figure and the caption to clarify these points and improve the line visibility for better understanding. Please refer to the revised manuscript for these changes.
>
> > **How can the use of the Gaussian filter and the TV proximity operator ensure effective decomposition in this method?**
>
> The reviewer raises an important point regarding the use of the Gaussian filter and the total variation (TV) proximity operator in the decomposition process.
>
> The Retinex decomposition can be perform by focusing on illumination estimation only. However, the locality and translation equivariance of covolutional layer is not insufficient for decomposing an image into piecewise smooth illumination. We therefore incoporate the TV optimization's edge-preserving properties into a single convolutional layer.
>
> Directly integrating a TV layer into the convolutional framework, however, results in inaccurate illumination estimation, as shown in Fig. 3. To address this, we replace the standard convolutional kernel with a fixed Gaussian kernel. This structured initialization stabilizes the illumination estimation process by leveraging the Gaussian kernel's consistent and smooth bias, ensuring reliable and edge-preserving outcomes.
>
> > **Why is gamma correction applied after estimating illumination, and why is the reflectance obtained only after applying gamma correction?**
>
> - **Gamma correction is applied after estimating the illumination** ($I$) to adjust its brightness, resulting in a corrected illumination ($I^\gamma$). We've added an example of the corrected illumination in Figure 4, where we can see the adjustment in the illumination distribution.
>
> - **It is the corrected reflectance, not the raw reflectance, that we use as the initial enhanced image.** The corrected reflectance is computed after gamma correction, using the formula $R = S \oslash I^\gamma$. We've included an example of this corrected reflectance in Figure 4. In contrast, the raw reflectance ($R = S \oslash I$) is not suitable for this purpose. The corrected reflectance is used to guiding the DDIM sampling process, ensuring the consistentcy of image content and structure.
>
> > **The values for the gamma correction factor in Equation (4) and the $\lambda$ in Equation (3) are not provided.**
>
> We have moved the experimental setups from appendix to the main manuscript in this revised version, as is highted in blue in Section 4.
>
> - The value of the Gamma factor $\gamma$ in Equation (4) is set to 0.5. The reason that we set value of $\gamma$ to 0.5 is as follow. The default gamma correction value for most display devices, such as computer monitors, is set to 2.2, this value provides a good balance between visual perception and device compatibility. Therefore, we set $\gamma = 1/ 2.2$ to adjust the illumination. For convience, we set $\gamma = 1/ 2 = 0.5$.
> - The value of $\lambda$ in Equation (3) is set to 30. In addition, the value of $\sigma$ in Equation(3) is expericaly set to 0.5.
>
>
> > **Does the illumination shown in Figure 3 represent the illumination before or after the gamma correction?**
>
> Thanks for your thoughtful comment. First, the illumination needs to be corrected using gamma correction to obtain the corrected illumination. With this corrected illumination, we can then compute the corrected reflectance. The corrected reflectance serves as the initial enhanced image and used to conditioning the sampling process.
>
> To improve clarity, we have updated Figure 4 to include both the pre-correction and post-correction versions of the illumination.

---

> ### Author Response · Authors · 2024-11-21
>
> > **...the reflectance appears over-saturated,... I am unsure whether this over-saturation is an effect of integrating gamma correction.**
>
> We apologize for the confusion. However, the reflectance is not utilized for subsequent personalized enhancement. Instead, we use the **corrected reflectance**, obtained by $R=S \oslash I^\gamma$, for subsequent personalized enhancement. The corrected reflectance is not over-saturated and can be regarded as initial enhanced image. The corrected reflectance is further used to conditioning the sampling process, benefiting ensuring the the consistentcy of image content and structure.
>
> We have updated Figure 4 to include both the reflectance and corrected reflectance for better clarity.
>
> > **It is unclear which components are fixed and which are learnable.  I hope authors provide a diagram of the gradient íow for clarity.**
>
> Thanks for your comments. Actually, our model is training-free, the weights of Gaussian convolutional layer are fixed. Both of the pretrained parameters of DDIM model and CLIP model are frozen. During the sampling process, the loss functions adjust the output by **adding gradients to the denoised image at each time step**. These gradients do not update model parameters but shift the mean of the unconditional distribution. This shift ensures the generated images during DDIM sampling align closely with the distribution of personalized augmented images while preserving content consistency. We encourage the reviewer to refer to Ref. [1] for further details.
>
> We have improved the overview in Figure 1 to clarify our method.
>
> >**The proposed method does not demonstrate enhanced performance compared to RetinexDIP.**
>
> We acknowledge that our method does not yet outperform RetinexDIP across all datasets. However, we would like to highlight that our model outperforms RetinexDIP on the LOL and VELOL datasets. We additionaly compare our model with RetinexDIP using NIQE, QIQMC and CPCQI, to demonstrate our model's superior performance, as shown in Table 3.
>
> >**..., the differences between results with different text instructions appear minimal, as illustrated by the bottom line in Figure 5.**
>
> You have raised an important point that the content and structures of enhanced image is consistently preserved well.  Regarding to the variation of styles, we can easily change the style of enhanced image by tuning the weight of $L_{clip}$, as demonstrated in Figure 14 in Appendix A.5.3. We have provided additional examples to show how the model responds to different text prompt.

---

> > ### Comment · Reviewer_2CJL · 2024-11-27
> >
> > Thanks for the detailed response from authors. Most of my concerns regarding the paper's writing have been addressed. However, based on Table 1, it appears that the proposed RetinexGDP performs worse than RetinexDIP on seven datasets in most cases. While performance is not the sole criterion for a paper's acceptance, being outperformed by a study published in 2022 raises concerns about the broader contribution of this paper to the research community. Furthermore, even after the rebuttal, there is still room for improvement in the overall quality of the paper. For instance, in Table 1, the N column of the DICM dataset contains two highlighted blue numbers, whereas the M column of the VV dataset has none. Based on the above analysis, I have decided to maintain my original rating.

---

> > > ### Author Response · Authors · 2024-11-27
> > >
> > > We appreciate your thoughtful comments and your careful review of our work.
> > >
> > > >Most of my concerns regarding the paper's writing have been addressed.
> > >
> > > We are glad that most of your concerns have been addressed. We appreciate your careful review and are committed to further refining the manuscript based on your valuable suggestions.
> > >
> > > >RetinexGDP performs worse than RetinexDIP...
> > > - **Performance Comparison with RetinexDIP:** We acknowledge that Table 1 shows instances where RetinexGDP does not outperform RetinexDIP on some datasets. However, we would like to emphasize that our RetinexGDP model was primarily evaluated on two critical datasets—LOL and VELOL—where it achieves superior performance, as detailed in Table 2:
> > >
> > > | Method          | LOL P$\uparrow$ | LOL S$\uparrow$ | LOL N$\downarrow$ | LOL M$\uparrow$ | LOL C$\uparrow$ | VELOL P$\uparrow$ | VELOL S$\uparrow$ | VELOL N$\downarrow$ | VELOL M$\uparrow$ | VELOL C$\uparrow$ |
> > > |-----------------|-----------------|-----------------|-------------------|-----------------|-----------------|-------------------|-------------------|---------------------|-------------------|-------------------|
> > > | **Zero-DCE**    | _14.86_     | 0.54        | 7.77          | 4.01        | 1.15        | **18.06**   | _0.58_      | 8.06          | 3.92        | 1.20        |
> > > | **CLIP-LIT**    | 12.39       | 0.49        | 8.29          | 3.37        | **1.21**    | 15.18       | 0.53        | 8.41          | 3.37        | **1.27**    |
> > > | **RetinexDIP**  | 8.59        | 0.30        | 6.90          | 2.41        | 1.10        | 11.08       | 0.32        | _7.23_        | 2.65        | 1.10        |
> > > | **NeuralBR**    | 11.36       | 0.44        | 7.52          | 3.68        | _1.17_      | 14.04       | 0.47        | 7.56          | 3.67        | _1.22_      |
> > > | **GDP**         | 13.93       | _0.63_      | **6.17**      | **5.34**    | 0.67        | 13.04       | 0.55        | 7.59          | _4.29_      | 0.40        |
> > > | **RetinexGDP**  | **15.66**   | **0.66**    | _6.26_        | _5.26_      | 0.85        | _16.51_     | **0.69**    | **6.92**      | **4.97**    | 0.96        |
> > >
> > > - **Broader Contribution of RetinexGDP:** In addition to performance improvements on specific datasets, RetinexGDP introduces a significant advancement in user-guided enhancement. Unlike previous methods, our approach allows for personalized enhancement, where users can specify their preferred enhancement styles through text. This level of flexibility in user interaction is a key distinguishing factor that sets RetinexGDP apart from existing methods, including RetinexDIP.
> > >
> > > - **Clarification of Table 1:** We have revised the table to ensure that the dataset columns are consistently formatted and that the highlighting is done appropriately to reflect the correct performance comparisons across all datasets. A '-' indicates results are unavailable due to memory issues.
> > >
> > > Thank you again for your valuable feedback. We believe that these revisions will significantly enhance the clarity of our work and strengthen the presentation of our contributions.

---

### Official Review · Reviewer_5eq8 · 2024-11-04

**Soundness:** 3
**Presentation:** 3
**Contribution:** 2
**Rating:** 5
**Confidence:** 4

**Summary:**

This paper proposes a new model called RetinexGDP which leverages Retinex decomposition model and DDIM inversion to address the zero-shot text-guided low-light enhancement problem. Specifically, the light component is estimated using a total-variation tailored Gaussian convolutional layer, and the enhanced image is estimated with the help of patch-wise DDIM inversion to maintain content and structure consistency. Experiments on multiple datasets indicate that the proposed method achieves comparable performance with state-of-the-art methods in terms of noise suppression and enhancement quality.

**Strengths:**

++ The experiments take various datasets (specifically 9 datasets) into consideration, providing a thorough evaluation of the proposed model.

**Weaknesses:**

-- My main concern is the limited novelty of the paper. The proposed method combines Retinex composition with GDP, and enables textual control. Most of the used techniques come from existing work, e.g., GDP, Retinex model, patch-wise DDIM inversion. The introduced total variation layer borrows the idea of Yeh et al. (2022). The proposed idea is not that inspiring.

-- The effectiveness of the model is not impressive. The model does not achieve state-of-the-art performance compared to other methods. Also, the proposed model does not achieve consistent performance across different metrics, as seen from Table 1, 2, and 3. Besides, the qualitative examples shown in Figure 5, 12, and 15 indicate that the proposed method could suffer from poor fidelity (some enhanced images have unnatural colors and styles (e.g., Figure 5) while some of them have over-smoothed effect (e.g., Figure 15)).

-- The paper proposes to replace the original total variation layer with fixed Guassian initialized kernels. I wonder whether the layer can still be called "total variation layer" or not. I think it has nothing to do with the "total variation layer".

-- The introduced patch-wise DDIM inversion has high computational complexity, which makes the proposed method less appealing and hinders possible application.

-- The textual control ability and granularity has not been fully demonstrated.

**Questions:**

1. Please further elaborate the proposed Guassian convolutional layer, and discuss its significance with thet original total variation layer.

2. Please futher explain how to specifically ensure content consistency across different textual prompts. More examples would be helpful.

---

> ### Author Response · Authors · 2024-11-21
>
> We appreciate your time and detailed feedback. We acknowledge the concerns raised and would like to clarify key aspects of our paper in order to address the perceived limitations and further highlight the contributions of our work.
>
>
> >**...the used techniques, ... total variation layer borrows the idea of Yeh et al. (2022)....**
>
>
> While our work draws inspiration from Yeh et al. (2022), the novelty lies in how we extend these ideas to tackle a new challenge: zero-shot, text-guided, personalized low-light image enhancement. Our contributions are:
>
> - **Innovative use of the TV optimization for zero-shot illumination estimation.** Previous zero-shot enhancement model, such as RetinexDIP (TCSVT 2021), exploiting DIP's inductive bias for Retinex decomposition, requires multiple deep networks and additional hand-crafted priors. Here, we incorporate the edge-preserving properties of TV optimization into a single convolutional layer, for for zero-shot illumination estimation, which has not been fully explored before.  In the original work by Yeh et al. (2022), the TV optimization as a layer is used to build a **deep neural network**, requiring **training** on paired dataset. In this paper, our illumination estimation requires **no training** process, and a single image is enough.
> - **Guiding generative diffusion prior (GDP) with text for personalized enhancement** Existing GDP based enhanced models face challenges with personalization and content consistency. We address this through several innovations:  **(i)** Conditioning the DDIM sampling process on the corrected reflectance rather than the low-light image, as in prior work. **(ii)** Using patch-wise DDIM inversion to initialize with the corrected reflectance, preserving image structures and textures. **(iii)** Adopting a patch-based strategy in DDIM inversion to optimize the initial noise vector. **(iv)** Guding DDIM sampling using text prompts, enabling flexible enhancement by specifying the preferences with text.
>
> By combining these contributions, our work is the first, to our knowledge, to achieve **zero-shot, text-based personalized low-light image enhancement**. Users can now specify enhancement styles via text instructions, making this a meaningful advancement in the field.
>
>
>
>
> >**...does not achieve state-of-the-art performance compared to other methods ...**
>
> We thank the reviewer for evaluating our model's performance. While we acknowledge that our method does not yet surpass all state-of-the-art models across all metrics, **it shows clear advantages in specific comparisons**. As shown in Table 2, compared to training-based models like CLIP-LIT (CVPR '23), our model achieves significantly higher PSNR scores: 26.39% on LOL and 8.7% on VELOL. Similarly, against training-free models such as RetinexDIP (TCSVT 2021), RetinexGDP delivers substantial improvements, with 82.3% higher PSNR on LOL and 48.9% on VELOL.
>
>
> >**The paper proposes to replace the original total variation layer with fixed Guassian initialized kernels...**
>
>
> We apologize for the confusion in the initial presentation and would like to clarify the design of the TV layer .
>
> Our approach does not replace the original total variation (TV) layer with a fixed Gaussian-initialized kernel. Instead, we incoporate the TV layer's edge-preserving properties into a single convolutional layer. Furthermore, we replace the **standard convolutional kernel** with a Gaussian-initialized kernel.
>
> >**... patch-wise DDIM inversion has high computational complexity....**
>
> We agree that our RetiexGDP has limitation in real-time enhancement. However, compared to GDP(CVPR '23), our RetinexGDP requires significantly fewer time steps and has a shorter running time, as demonstrated in below Table.
> The visulization of denoising process in Figure 10 and 12 demonstrate that our RetinexGDP can give pleasing result within 10 iterations, in contrast to 1000 iterations in GDP.
>
> |  | GDP(time step=1000) | RetinexGDP(time step=25) |
> | :---: |  :---: | :---:  |
> | Time | 20 min  |  6 min |
> | PSNR(VELOL) |   13.04|  16.51|

---

> ### Author Response · Authors · 2024-11-21
>
> >**The textual control ability and granularity has not been fully demonstrated.**
>
> We appreciate the reviewer's suggestion to further demonstrate the granularity and effectiveness of the textual control ability. In the revised version of the paper, we have included additional examples that illustrate how the model's output varies with different textual prompts, as shown in Figure 6. Our RetinexGDP enables enabling enhance low-light image by freely specifying the preferred styles using texts.
>
>
> >**Please further elaborate the proposed Guassian convolutional layer, and discuss its significance with the original total variation layer.**
>
> We aim to perform zero-shot illumination estimation. We therefore incoporate the TV layer's edge-preserving properties into a single convolutional layer. Directly integrating a TV layer into the convolutional framework, however, results in inaccurate illumination estimation, as shown in Fig. 3(a). This issue arises because without parameter updates via backpropagation, the convolutional layer's weights remain random, leading to incorrect image enhancement outcomes.
>
> To address this, we **replace the standard convolutional kernel with a fixed  Gaussian kernel**. The weights of Gaussian is fixed, will no be updated. This  Gaussian TV layer stabilizes the illumination estimation process, ensuring reliable and edge-preserving outcomes.
>
>
> > **Please futher explain how to specifically ensure content consistency across different textual prompts. More examples would be helpful.**
>
> The content consistency across different textual prompts is ensured by following contributions:
>
> - **Patch-wise strategy is adopted in DDIM inversion.** The initial noise vector plays a crucial role in maintaining the fidelity of the generated original image. In the inversion process, the corrected reflectance is first divided into M overlapping patches. For each patch, we obtain the noised intermediate result. Since the patches overlap, the overlapping areas are computed multiple times. This approach ensures that the contributions from overlapping patches are aggregated, enabling generate a good initial noise vector. This initial noise vector benefits preserving the consistency of the image's structure and content during the subsequent sampling process.
> - **Conditioning the DDIM sampling process with corrected reflectance.** Starting from the initial noise vector, the sampling process is conditioned with corrected reflectance. The loss function, $L_{recon}$ and $L_{per}$ ensure that the content and structure of generated enhanced image is consistent with the corrected reflectance (Again, the corrected reflectance, rather than the reflectance, is used as guidance. A corrected reflectance can be regarded as the initial enhanced image).
>
> More examples have been provided in Figure 6.

---

### Author Response · Authors · 2024-11-27
**Follow-up on Our Response**

Dear ICLR Reviewers,

Thank you for taking the time to review my submission. I am writing to follow up on my response to the reviewer comments, which I submitted a few days ago. As I have not yet received any feedback, I would like to confirm that my response has been received and is under consideration. To reiterate, the key contributions of my paper are as follows:

**Zero-shot text-based personalized low-light enhancement model**: We propose a personalized low-light enhancement model called RetinexGDP, which enables flexible low-light image enhancement guided by user preferences specified via text instructions, all without the need for additional training or external images.

**Edge-aware Retinex decomposition**: We introduce an edge-aware total variation optimization property, integrated into a single Gaussian convolutional layer, to achieve zero-shot Retinex decomposition.

**Patch-wise DDIM inversion**: We employ patch-wise DDIM inversion to generate the initial noise vector for corrected reflectance and use the corrected reflectance as a condition for the DDIM reverse process.

If there are any further clarifications or additional information needed regarding my submission, I would be happy to provide more details. I look forward to your feedback.Thank you for your time and consideration.

---

### Note · Authors · 2025-03-06

I have read and agree with the venue's withdrawal policy on behalf of myself and my co-authors.

---

### Meta-Review · Area_Chair_77Lj · 2024-12-19

**Metareview:**

The paper presents RetinexGDP, a training-free, zero-shot personalized low-light image enhancement model that integrates Retinex theory with deep learning. The method leverages textual control to allow users to specify their preferences for image enhancements. It is designed to decompose images into illumination and reflectance components, applying enhancements primarily to reflectance based on user-provided text instructions via the CLIP model.

***Strengths:***
- The integration of Retinex information and the CLIP model enhances the interpretability and flexibility of the enhancement process.
- The model is evaluated across a diverse set of nine datasets.
- The method is training-free and zero-shot, which could potentially reduce training burden.

***Weaknesses:***
- The core techniques employed, such as GDP, Retinex model, and patch-wise DDIM inversion, are largely derived from existing works.
- The proposed total variation layer modification lacks originality and is questioned for its relevance to traditional TV.
- The effectiveness of the model is not consistently superior to state-of-the-art methods, and issues with fidelity and unnatural effects in enhanced images are noted.
- Several technical details, such as parameters and the operational specifics of components like Gaussian filters and gamma correction, are insufficiently explained or justified.
- The effectiveness and granularity of textual control over the image enhancement process are not convincingly demonstrated.

We appreciate the authors' efforts in submitting their rebuttal and addressing some of the technical details. However, the fundamental concerns about novelty and performance remain unresolved.Given these critical issues wrt limited novelty and permance advantage, the manuscript does not meet the high standards required for acceptance in its current form.

**Additional Comments On Reviewer Discussion:**

The authors submitted a rebuttal that included additional explanations and comparisons with methods such as LightenDiffusion and FourierDiff. However, concerns regarding the novelty of the approach and the insufficient performance gains remain significant. Despite the rebuttal, three reviewers maintained their negative evaluations. The authors were unable to convincingly address their concerns. The prevailing issues with novelty and performance ultimately lead to the decision to reject.

---

### Decision · Program_Chairs · 2025-01-22

Reject